# METTL18-mediated histidine methylation of RPL3 modulates translation elongation for proteostasis maintenance

Eriko Matsuura-Suzuki[1†], Tadahiro Shimazu[2\*†], Mari Takahashi[3], Kaoru Kotoshiba[2], Takehiro Suzuki[4], Kazuhiro Kashiwagi[3], Yoshihiro Sohtome[5,6], Mai Akakabe[6], Mikiko Sodeoka[5,6], Naoshi Dohmae[4], Takuhiro Ito[3\*], Yoichi Shinkai[2\*], Shintaro Iwasaki[1,7\*]

[1]RNA Systems Biochemistry Laboratory, RIKEN Cluster for Pioneering Research, Saitama, Japan; [2]Cellular Memory Laboratory, RIKEN Cluster for Pioneering Research, Saitama, Japan; [3]Laboratory for Translation Structural Biology, RIKEN Center for Biosystems Dynamics Research, Yokohama, Japan; [4]Biomolecular Characterization Unit, Technology Platform Division, RIKEN Center for Sustainable Resource Science, Saitama, Japan; [5]RIKEN Center for Sustainable Resource Science, Saitama, Japan; [6]Synthetic Organic Chemistry Lab, RIKEN Cluster for Pioneering Research, Saitama, Japan; [7]Department of Computational Biology and Medical Sciences, Graduate School of Frontier Sciences, The University of Tokyo, Chiba, Japan

**\*For correspondence:**
tshimazu@riken.jp (TS);
takuhiro.ito@riken.jp (TI);
yshinkai@riken.jp (YS);
shintaro.iwasaki@riken.jp (SI)

[†]These authors contributed equally to this work

**Abstract** Protein methylation occurs predominantly on lysine and arginine residues, but histidine also serves as a methylation substrate. However, a limited number of enzymes responsible for this modification have been reported. Moreover, the biological role of histidine methylation has remained poorly understood to date. Here, we report that human METTL18 is a histidine methyltransferase for the ribosomal protein RPL3 and that the modification specifically slows ribosome traversal on Tyr codons, allowing the proper folding of synthesized proteins. By performing an in vitro methylation assay with a methyl donor analog and quantitative mass spectrometry, we found that His245 of RPL3 is methylated at the $\tau$-N position by METTL18. Structural comparison of the modified and unmodified ribosomes showed stoichiometric modification and suggested a role in translation reactions. Indeed, genome-wide ribosome profiling and an in vitro translation assay revealed that translation elongation at Tyr codons was suppressed by RPL3 methylation. Because the slower elongation provides enough time for nascent protein folding, RPL3 methylation protects cells from the cellular aggregation of Tyr-rich proteins. Our results reveal histidine methylation as an example of a ribosome modification that ensures proteome integrity in cells.

## Editor's evaluation

This study investigates METTLL18-mediated RPL3 histidine methylation on 245 position and how it regulates translation elongation and protects cells from cellular aggregation of Tyr-rich proteins. The study potentially provides some new example of 'ribosome code' and how ribosome PTM could affect protein translation.

## Introduction

Protein methylation is an integral post-translational modification (PTM) that contributes to the critical role of epigenetics. This modification influences the function of proteins and provides a convertible

platform for the modulation of cellular processes, including interactions with other molecules, protein structure, localization, and enzymatic activity (*Biggar and Li, 2015*; *Clarke, 2018*; *Murn and Shi, 2017*; *Rodríguez-Paredes and Lyko, 2019*). Whereas the majority of protein methylation has been observed on lysine and arginine amino acids (*Biggar and Li, 2015*; *Clarke, 2018*; *Murn and Shi, 2017*; *Rodríguez-Paredes and Lyko, 2019*), histidine residues also supply alternative target sites for methylation. Although this modification was previously thought to be restricted to limited proteins (*Webb et al., 2010*), a recent comprehensive survey suggested widespread occurrence in the proteome (*Davydova et al., 2021*; *Ning et al., 2016*; *Wilkinson et al., 2019*).

Two distinct nitrogen atoms in histidine could be methylated: the π-*N* position (π-*N*-methylhistidine or 1-methylhistidine) and the $\tau$-*N* position ($\tau$-*N*-methylhistidine or 3-methylhistidine) (*Figure 1— figure supplement 1A*). Regarding their responsible enzymes, methyltransferase-like (METTL) 9 – a seven β-strand methyltransferase – and SET domain containing 3 (SETD3) are, as of yet, the only known mammalian protein methyltransferases for π-*N*-methylhistidine (*Davydova et al., 2021*) and $\tau$-*N*-methylhistidine (*Dai et al., 2019*; *Guo et al., 2019*; *Kwiatkowski et al., 2018*; *Wilkinson et al., 2019*; *Zheng et al., 2020*), respectively. Nevertheless, the landscape of histidine methyltransferase-substrate pairs and, more importantly, the physiological functions of the modification have remained largely elusive.

Recent emerging studies have shown that the ribosome is a hotspot for PTM (*Emmott et al., 2019*; *Simsek and Barna, 2017*). This led to the idea of gene regulation through the decorated ribosomes. Indeed, PTMs on ribosomal proteins may regulate protein synthesis in specific contexts, such as a subset of transcripts (*Imami et al., 2018*; *Kapasi et al., 2007*; *Mazumder et al., 2003*), the cell cycle (*Imami et al., 2018*), stress response (*Higgins et al., 2015*; *Matsuki et al., 2020*), and differentiation (*Werner et al., 2015*).

In this work, we studied the previously uncharacterized methyltransferase METTL18. Through a survey of the substrate, we found that this protein catalyzes $\tau$-*N*-methylation on His245 of the ribosomal protein large subunit (RPL) 3 (uL3 as universal nomenclature). Cryo-electron microscopy (cryo-EM) suggested that $\tau$-*N*-methylation interferes with the interaction of His245, which is located close to the peptidyl transferase center (PTC), with G1595 in the loop of helix 35 in 28S rRNA. Genome-wide ribosome profiling revealed that the $\tau$-*N*-methylhistidine on RPL3 retards translation elongation at Tyr codons, a conclusion that was also supported by an in vitro translation assay. Quantitative proteome analysis showed that slower elongation is associated with the proper folding of synthesized proteins and thus ensures healthy proteostasis in cells; otherwise, proteins are aggregated and degraded. Our study provided an example of a modified ribosome as a nexus for the quality control of synthesized protein.

## Results

### METTL18 is a histidine τ-*N*-methyltransferase

To overview $\tau$-*N*-histidine methylation in cells and the corresponding enzymes, we quantified $\tau$-*N*-methylhistidine by mass spectrometry (MS). To distinguish the two types of histidine methylations, we performed multiple reaction monitoring (MRM) of digested amino acids to trace specific *m/z* transitions from the precursor (*Davydova et al., 2021*) (see 'Materials and methods' section for details). Because SETD3 modifies abundant actin proteins (*Dai et al., 2019*; *Guo et al., 2019*; *Kwiatkowski et al., 2018*; *Wilkinson et al., 2019*; *Zheng et al., 2020*), *SETD3* knockout (KO) in HEK293T cells (*Figure 1—figure supplement 1B–E*) greatly reduced $\tau$-*N*-methylhistidine (*Figure 1A*). However, a substantial fraction of $\tau$-*N*-methylhistidine was left in the *SETD3* KO cells (*Figure 1A*), suggesting the presence of other mammalian $\tau$-*N*-methyltransferase(s).

An apparent candidate for the remaining $\tau$-*N*-methyltransferase in *SETD3* KO cells was METTL18, whose yeast homolog (histidine protein methyltransferase 1 [Hpm1]) has been reported to catalyze $\tau$-*N*-methylation on histidine in Rpl3 (*Al-Hadid et al., 2016*; *Al-Hadid et al., 2014*; *Webb et al., 2010*). Indeed, the *SETD3-METTL18* double-KO (DKO) cells (*Figure 1—figure supplement 1B–E*) had even lower levels of $\tau$-*N*-methylhistidine than the *SETD3* single-KO cells (*Figure 1A*). Note that we could not detect any significant alteration in π-*N*-methylhistidine in any cells tested in this study (*Figure 1—figure supplement 1F*).

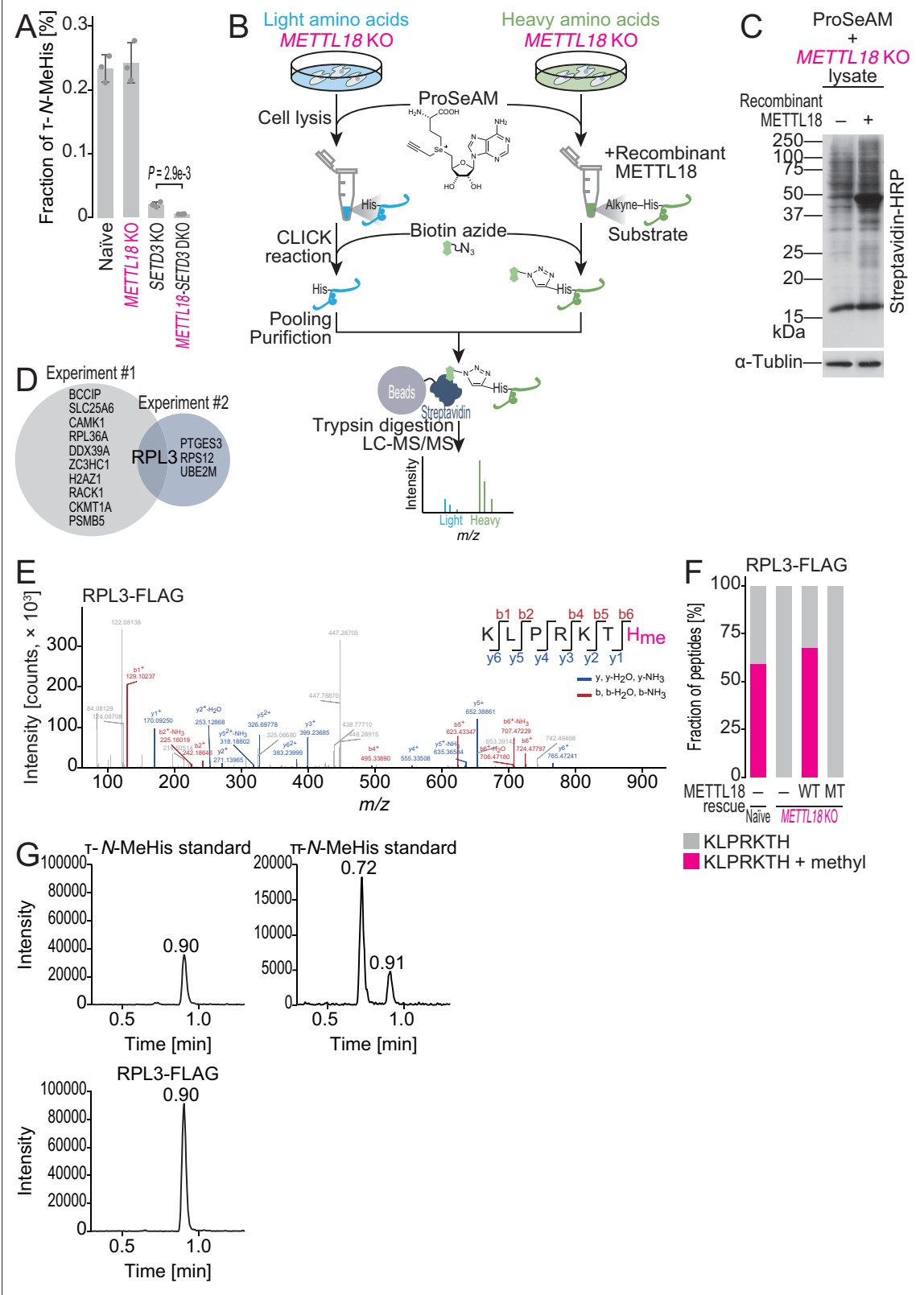

**Figure 1.** ProSeAM-SILAC identifies RPL3 as a substrate of METTL18. (**A**) Multiple reaction monitoring (MRM)-based identification of $\tau$-$N$-methylated histidine in bulk proteins from the indicated cell lines. Data from three replicates (points) and the mean (bar) with SD (error bar) are shown. Significance was determined by Student's $t$-test (unpaired, two-sided). (**B**) Schematic representation of the ProSeAM-SILAC approach. (**C**) ProSeAM-labeled proteins in cell lysate with recombinant His-METTL18 protein. Biotinylated proteins were detected by streptavidin-HRP. Western blot for α-tubulin was used

*Figure 1 continued on next page*

*Figure 1 continued*

as a loading control. (**D**) Venn diagram of proteins identified in two independent ProSeAM-SILAC experiments. The reproducibly detected protein was RPL3. (**E**) Methylated histidine residue in ectopically expressed RPL3-FLAG was searched by liquid chromatography mass spectrometry (LC-MS/MS). (**F**) Quantification of methylated and unmethylated peptides (KLPRKTH) from the indicated cells. RPL3-FLAG was ectopically expressed and immunopurified for LC-MS/MS. WT, wild type; MT, Asp193Lys-Gly195Arg-Gly197Arg mutant. (**G**) MRM-based identification of $\tau$-*N*-methylhistidine in peptides from RPL3. The $\tau$-*N*-methylhistidine standard, π-*N*-methylhistidine standard, and RPL3-FLAG peptide (KLPRKTH) results are shown. MeHis, methylhistidine.

The online version of this article includes the following source data and figure supplement(s) for figure 1:

**Source data 1.** Full and unedited blots corresponding to *Figure 1C*.

**Source data 2.** Primary data for graphs in *Figure 1*.

**Figure supplement 1.** Generation of *SETD3* and *METTL18* knockout (KO) cells.

**Figure supplement 1—source data 1.** Full and unedited blots corresponding to *Figure 1—figure supplement 1C*.

**Figure supplement 1—source data 2.** Full and unedited blots corresponding to *Figure 1—figure supplement 1D*.

**Figure supplement 1—source data 3.** Full and unedited gel images corresponding to *Figure 1—figure supplement 1G*.

**Figure supplement 1—source data 4.** Primary data for graphs in *Figure 1—figure supplement 1E and F*.

**Figure supplement 2.** Characterization of methylhistidine in endogenous RPL3.

**Figure supplement 2—source data 1.** Full and unedited gel images corresponding to *Figure 1—figure supplement Figure 1—figure supplement 2B*.

**Figure supplement 2—source data 2.** Primary data for graphs in *Figure 1—figure supplement 2D*.

## METTL18 catalyzes τ-*N*-methylation on His245 in RPL3

These observations led us to survey the methylation substrate of METTL18. For this purpose, we harnessed propargylic *Se*-adenosyl-ʟ-selenomethionine (ProSeAM), an analog of *S*-adenosyl-ʟ-methionine (SAM). This compound acts as a donor in the methylation reaction by the SET domain and 7-β-strand methyltransferase (*Davydova et al., 2021*; *Shimazu et al., 2014*; *Shimazu et al., 2018*). Instead of a methyl moiety, a propargyl unit was added to the substrate residue, allowing biotin tagging with a click reaction (*Davydova et al., 2021*; *Shimazu et al., 2014*; *Shimazu et al., 2018*; *Figure 1B*). Using ProSeAM, we performed in vitro methylation with recombinant METTL18 (*Figure 1—figure supplement 1G*) in the lysate of *METTL18* KO cells and detected the ProSeAM-reacted proteins in a METTL18 protein-dependent manner (*Figure 1C*). The high background signals are likely to have originated from methyltransferases other than METTL18 in the lysate.

For the quantitative and sensitive proteomic identification of methylated protein(s), we combined this approach with stable isotope labeling using amino acids in cell culture (SILAC) (*Figure 1B*). ProSeAM was added to the cell lysates of *METTL18* KO cells labeled with either light or heavy isotopic amino acids. Only the extract from heavy-isotope-labeled cells was incubated with recombinant METTL18 (*Figure 1B*). After lysate pooling and streptavidin purification (*Figure 1B*), isolated proteins were quantitatively assessed by liquid chromatography (LC)-MS/MS. Among the detected proteins, human RPL3 was the only candidate to be reproducibly identified (*Figure 1D*).

To ensure cellular histidine methylation on the protein, FLAG-tagged RPL3 was expressed in naïve HEK293T cells, immunopurified, and subjected to LC-MS/MS. This analysis identified the methylhistidine in cellular RPL3 and precisely annotated the residue at His245 (*Figure 1E*). In stark contrast, the same experiments with *METTL18* KO did not detect methylated His245 (*Figure 1F*). The ectopic expression of wild-type (WT) METTL18 in the KO cells rescued the modification, whereas mutations in the potential SAM binding site (Asp193Lys-Gly195Arg-Gly197Arg), which were predicted based on lysine methyltransferase orthologs (*Ng et al., 2002*), abolished the potential (*Figure 1F*). The same METTL18-dependent His245 methylation was also found in endogenous RPL3 (*Figure 1—figure supplement 2C and D*), isolated in the 60S subunit (*Figure 1—figure supplement 2A and B*).

To distinguish the methylation forms on histidine, we applied MRM to the peptide fragments containing His245 and clearly observed that methylation occurs at the $\tau$-*N* position but not at the π-*N* position (*Figure 1G*).

Thus, taken together, our data demonstrated that His245 of RPL3 is a METTL18 substrate for the formation of $\tau$-*N*-methylhistidine in cells.

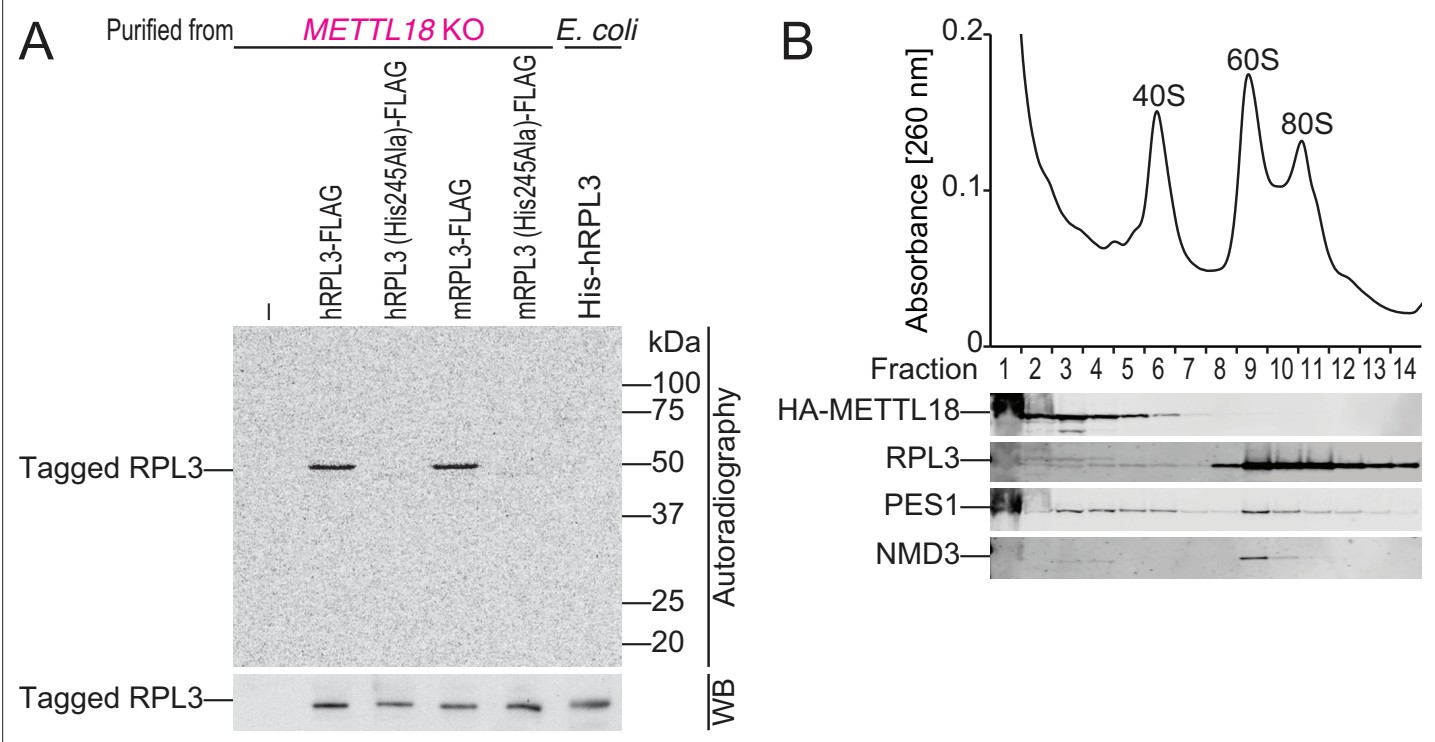

**Figure 2.** METTL18 associates with pre-60S. (**A**) In vitro methylation assay with recombinant His-GST-METTL18 protein and $^{14}$C-labeled S-adenosyl-L-methionine (SAM). Immunopurified human or mouse RPL3 expressed in *METTL18* knockout (KO) cells and recombinant human RPL3 expressed in bacteria were used as substrates. (**B**) Western blot for the indicated proteins in ribosomal complexes separated by sucrose density gradient.

The online version of this article includes the following source data for figure 2:

**Source data 1.** Full and unedited images corresponding to *Figure 2A*.

**Source data 2.** Full and unedited blots corresponding to *Figure 2B*.

## METTL18 associates with early pre-60S

As RPL3 in crude lysate was methylated in vitro (*Figure 1C and D*), we set out to recapitulate this reaction by purified factors using $^{14}$C-labeled SAM as a methyl donor. Irrespective of the human or mouse homolog, the immunopurified FLAG-tagged RPL3 proteins transiently expressed in *METTL18* KO cells were efficiently labeled by recombinant METTL18 (*Figure 2A*). Moreover, changing His245 to Ala completely abolished the reaction, validating His245 as a methylation site.

In contrast, recombinant RPL3 expressed in bacteria was a poor substrate (*Figure 2A*). A similar deficiency in the in vitro methylation assay by Hpm1 on solo RPL3 protein produced in bacteria was seen in yeast (*Al-Hadid et al., 2014*). Given that RPL3 expressed in mammalian cells assembled into ribosomes but RPL3 expressed in bacteria did not, we hypothesized that METTL18 recognizes RPL3 within ribosomes or preassembled intermediates. To characterize the molecular complex associated with METTL18, we separated the ribosomal complexes through a sucrose density gradient and found that METTL18 was associated with a complex smaller than mature 60S (*Figure 2B*). Given the smaller size, we speculated that the METTL18-associating complex is pre-60S in the middle of biogenesis. Indeed, the METTL18-containing subfractions also possessed PES1 (a yeast Nop7 homolog), a ribosome biogenesis factor at an early step (*Kater et al., 2017*; *Sanghai et al., 2018*), and a small portion of RPL3 (*Figure 2B*). On the other hand, NMD3, an adaptor protein of the late-stage pre-60S for cytoplasmic export (*Ma et al., 2017*; *Malyutin et al., 2017*), was almost exclusive to the METTL18-associating pre-60S. These data suggested that RPL3, in an early intermediate complex of 60S biogenesis, is an efficient substrate for METTL18.

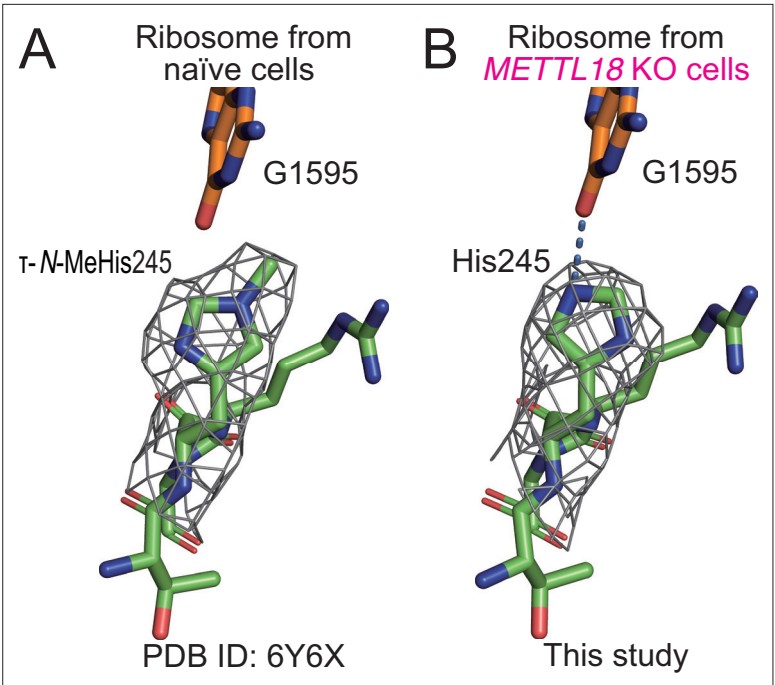

**Figure 3.** Structural differences in ribosomes upon methylation at His245. (**A**) Stick models of [244]GHR[246] of RPL3 and G1595 of the 28S rRNA of the human ribosome are shown with the cryo-electron microscopy (cryo-EM) density map around His245. The $\tau$-$N$-methyl group was manually added to the original model (PDB ID: 6Y6X) (*Osterman et al., 2020*) based on the cryo-EM density map. (**B**) The same model as in (**A**) of human ribosome from *METTL18* knockout (KO) cells. A hydrogen bond between His245 and G1595 is indicated with a dotted blue line.

The online version of this article includes the following source data and figure supplement(s) for figure 3:

**Figure supplement 1.** Ribosome subunit ratio in *METTL18* cells.

**Figure supplement 1—source data 1.** Full and unedited blots corresponding to *Figure 3—figure supplement 1D*.

**Figure supplement 1—source data 2.** Primary data for graphs in *Figure 3—figure supplement 1C and F*.

**Figure supplement 2.** Characterization of the structure of the 60S subunit from *METTL18* knockout (KO) cells.

## Structural comparison of methylated and unmethylated His245 of RPL3 in ribosomes

The presence of METTL18 in the pre-60S complex led us to investigate the role of methylation in ribosome biogenesis. However, assessed by the bulk 28S rRNA abundance (*Figure 3—figure supplement 1A*) and the 60S fraction in sucrose density gradient (*Figure 3—figure supplement 1B and C*), no altered abundance of the 60S subunit (relative to the 40S subunit) was observed in *METTL18* KO cells, while the knockdown of RPL17, which is known to hamper 60S biogenesis (*Wang et al., 2015*), reduced the 60S abundance in a sucrose density gradient (*Figure 3—figure supplement 1D–F*).

Thus, we hypothesized that RPL3 methylation may impact protein synthesis. To understand the potential role of methylation in RPL3, we reanalyzed published cryo-EM data (*Osterman et al., 2020*) and assessed the $\tau$-$N$-methyl moiety on His245 of RPL3 (*Figure 3A*), which indicated that $\tau$-$N$-methylated RPL3 is a stoichiometric component of the ribosome. A similar density of methylation on the histidine of RPL3 has also been found in rabbit ribosomes (*Bhatt et al., 2021*). On the other hand, the ribosomes isolated from *METTL18* KO cells lost density at the $\tau$-$N$ position of the His245 (*Figure 3B*, *Figure 3—figure supplement 2*, *Table 1*). The absence of methylation at the $\tau$-$N$ position allowed nitrogen to form hydrogen bonds with G1595 of 28S rRNA (*Figure 3B*). Given that G1595 is located in the loop of helix 35, which macrolide antibiotics target in bacterial systems (*Kannan and Mankin, 2011*), this difference in the interaction between His245 and G1595 suggests an alteration in the translation reaction.

**Table 1.** Data collection, model building, refinement, and validation statistics for cryo-electron microscopy (cryo-EM) data obtained in this study.

| | Human large ribosomal subunit (obtained from *METTL18* KO cells) (PDB: 7F5S, EMD-31465) |
|---|---|
| **Data collection and processing** | |
| Microscope | Tecnai Arctica |
| Camera | K2 Summit |
| Magnification | 39,000 |
| Voltage (kV) | 200 |
| Electron exposure (e⁻/Å²) | 50 |
| Exposure per frame | 1.25 |
| Number of frames collected | 40 |
| Defocus range (μm) | −1.5 to −3.1 |
| Micrographs (no.) | 5,517 |
| Pixel size (Å) | 0.97 |
| 3D processing package | RELION-3.1 |
| Symmetry imposed | C1 |
| Initial particle images (no.) | 381,227 |
| Final particle images (no.) | 118,470 |
| Initial reference map | EMD-9701 (40 Å) |
| *RELION estimated accuracy* | |
| Rotations (°) | 0.162 |
| Translations (pixel) | 0.287 |
| *Map resolution* | |
| masked (FSC = 0.143, Å) | 2.72 |
| Map sharpening B-factor | −63.0 |
| **Refinement** | |
| Model refinement package | phenix.real_space_refine |
| Initial model used | 6QZP |
| *Model composition* | |
| Chains | 45 |
| Non-hydrogen atoms | 138,634 |
| Residues | Protein: 6509; nucleotide: 3991 |
| Ligands | ZN: 5, MG: 297 |

*Table 1 continued on next page*

*Table 1 continued*

| | Human large ribosomal subunit (obtained from *METTL18* KO cells) (PDB: 7F5S, EMD-31465) |
|---|---|
| *B factors (Å²)* | |
| Protein | 62.85 |
| Nucleotide | 81.15 |
| *r.m.s. deviations* | |
| Bond lengths (Å) | 0.010 |
| Bond angles (°) | 0.834 |
| *Validation* | |
| Molprobity score | 1.92 |
| Clashscore | 9.61 |
| Poor rotamers (%) | 0.13 |
| CaBLAM outliers (%) | 3.35 |
| *Ramachandran plot* | |
| Favored (%) | 93.79 |
| Allowed (%) | 6.08 |
| Disallowed (%) | 0.12 |
| Map CC (CCmask) | 0.90 |

## Methylation of His245 of RPL3 slows ribosome traverse at Tyr codons

To investigate the impacts of the modification on protein synthesis, we assessed global translation in *METTL18* KO cells. However, we could not detect a significant difference in overall translation probed by polysome formation (*Figure 4—figure supplement 1A and B*), whereas RPL17 knockdown reduced the polysome (*Figure 4—figure supplement 1C and D*). Similarly, we could not observe a significant alteration in nascent peptides labeled with *O*-propargyl-puromycin (OP-puro) (*Figure 4—figure supplement 1E*).

Nonetheless, we investigated the implications of RPL3 methylation defects across the transcriptome by ribosome profiling (*Ingolia et al., 2009*; *Iwasaki and Ingolia, 2017*). Strikingly, we found increased translation elongation of Tyr codons in *METTL18* KO cells; ribosome occupancy on Tyr codons at the A-site was selectively reduced in the mutant cells (*Figure 4A and B*). This trend in ribosome occupancy was not observed at the P and E sites (*Figure 4—figure supplement 2A and B*). To evaluate the amino acid context associated with the high elongation rate, we surveyed the motifs around the A-site with reduced ribosome

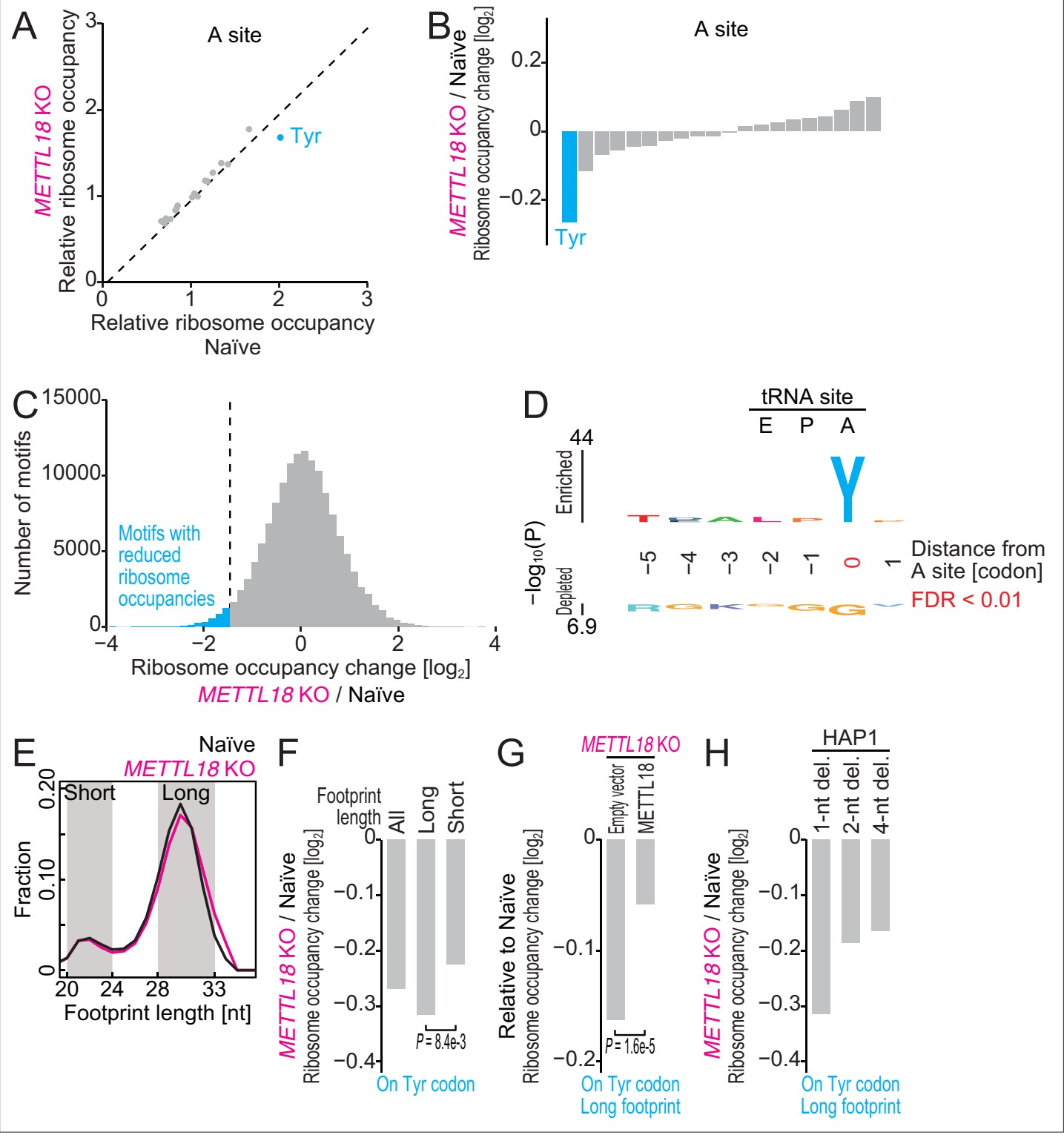

**Figure 4.** Ribosome profiling reveals Tyr codon-specific translation retardation by RPL3 methylation. (**A**) Ribosome occupancy at A-site codons in naïve and *METTL18* knockout (KO) HEK293T cells. Data were aggregated into codons with each amino acid species. (**B**) Ribosome occupancy changes at A-site codons caused by *METTL18* KO. (**C**) Histogram of ribosome occupancy changes in *METTL18* KO cells across motifs around A-site codons (seven amino acid motifs). Cyan: motifs with reduced ribosome occupancy (defined by ≤ mean − 2 SD). (**D**) Amino acid motifs associated with reduced ribosome occupancy in *METTL18* KO cells (defined in **C**) are shown relative to the A-site (at the 0 position). (**E**) Distribution of footprint length in naïve and *METTL18* KO HEK293T cells. (**F**) Ribosome occupancy changes on Tyr codons by *METTL18* KO along all, long (28–33 nt), and short (20–24 nt)

*Figure 4 continued on next page*

*Figure 4 continued*

footprints. Significance was determined by the Mann–Whitney *U*-test. (**G**) The recovery of long footprint reduction in *METTL18* KO cells by ectopic expression of METTL18 protein. Significance was determined by the Mann–Whitney *U*-test. (**H**) Changes in ribosome occupancy on Tyr codons by *METTL18* KO in HAP1 cells along long (28–33 nt) footprints. Del., deletion. In (**A–C**) and (**E–H**), the means of two independent experiments are shown.

The online version of this article includes the following source data and figure supplement(s) for figure 4:

**Figure supplement 1.** Basal translation activity in *METTL18* cells.

**Figure supplement 1—source data 1.** Primary data for graphs in *Figure 4—figure supplement 1B, D, and E*.

**Figure supplement 2.** Characterization of ribosome occupancy monitored by ribosome profiling.

**Figure supplement 2—source data 1.** Full and unedited blots corresponding to *Figure 4—figure supplement 2C*.

**Figure supplement 2—source data 2.** Full and unedited blots corresponding to *Figure 4—figure supplement 2E*.

**Figure supplement 2—source data 3.** Primary data for graphs in *Figure 4—figure supplement 2D and F*.

occupancy by METTL18 depletion (*Figure 4C*) and analyzed the enriched/depleted sequence in the group (*Figure 4D*). Remarkably, Tyr at the A-site was the predominant determinant for fast elongation in *METTL18* KO cells (*Figure 4D*).

The smoother elongation may be caused by increased tRNA abundance. However, tRNA$^{Tyr}_{GUA}$, which decodes both UAU and UAC codons, was noticeably reduced by METTL18 depletion (*Figure 4—figure supplement 2C and D*). On the other hand, the abundance control tRNA$^{Leu}_{HAG}$ was not altered (*Figure 4—figure supplement 2E and F*). Thus, tRNA abundance did not explain the reduced footprints on Tyr codons.

Further ribosome profiling data analysis supported that the accommodation of tRNA at the A-site may not explain the modulation of elongation by RPL3 methylation. Ribosome footprints possess two distinct populations of different lengths (short, peaked at ~22 nt; long, peaked at ~29 nt), reflecting the presence of A-site tRNA (*Lareau et al., 2014*; *Wu et al., 2019*; *Figure 4E*). Since A-site tRNA-free ribosomes are more susceptible to RNase treatment at the 3′ end, the trimmed short footprints represent nonrotated ribosomes waiting for A-site codon decoding by tRNA. On the other hand, long footprints originate from ribosomes accommodated with A-site tRNA in the middle of the peptidyl transfer reaction or subsequent subunit rotation (*Lareau et al., 2014*; *Wu et al., 2019*). The reduced footprints on Tyr codons in *METTL18* KO cells were more prominent in long footprints than short footprints (*Figure 4F*). Moreover, the long footprints were recovered by ectopic expression of METTL18 protein (*Figure 4G*). These data suggested that the process or processes downstream of codon decoding are promoted in KO cells. This observation was reasonable, given the relatively long distance between the $\tau$-*N*-methylated histidine and the mRNA codon.

To test the cell-type specificity, we further extended our ribosome profiling experiments in HAP1 cells, which originate from chronic myelogenous leukemia. We investigated ribosome footprints on the Tyr codons in three independent *METTL18* KO cell lines (1-nt, 2-nt, and 4-nt deletions) that should lead to truncated proteins (*Figure 4—figure supplement 2G*) and observed the reduction of ribosome occupancies on the codons in those cell lines as well (*Figure 4H*). These data suggested consistent effects of RPL3 methylation on elongation irrespective of cell type.

## Ribosomes deficient for RPL3 methylation exhibit higher processivity on Tyr codons in vitro

To further assess the effect of RPL3 methylation on Tyr codon elongation, we combined hybrid in vitro translation (*Erales et al., 2017*; *Panthu et al., 2015*) with the *Renilla*-firefly luciferase fusion reporter system (*Kisly et al., 2021*; *Figure 5A*). In this setup, we purified ribosomes from HEK293T (naïve or *METTL18* KO) cells, added them to ribosome-depleted rabbit reticulocyte lysate (RRL), and then conducted an in vitro translation assay (i.e., hybrid translation) (*Erales et al., 2017*; *Panthu et al., 2015*; *Figure 5A*). Indeed, we observed that removal of the ribosomes from RRL decreased protein synthesis in vitro and that complementation with ribosomes from HEK293T cells recovered the activity (*Figure 5—figure supplement 1A*).

To test the effect on Tyr codon elongation, we harnessed the fusion of *Renilla* and firefly luciferases; this system makes it possible to detect the delay/promotion of downstream firefly luciferase synthesis compared to upstream *Renilla* luciferase and thus to focus on the effect of the sequence inserted between the two luciferases on elongation (*Kisly et al., 2021*; *Figure 5A*). For better detection of the

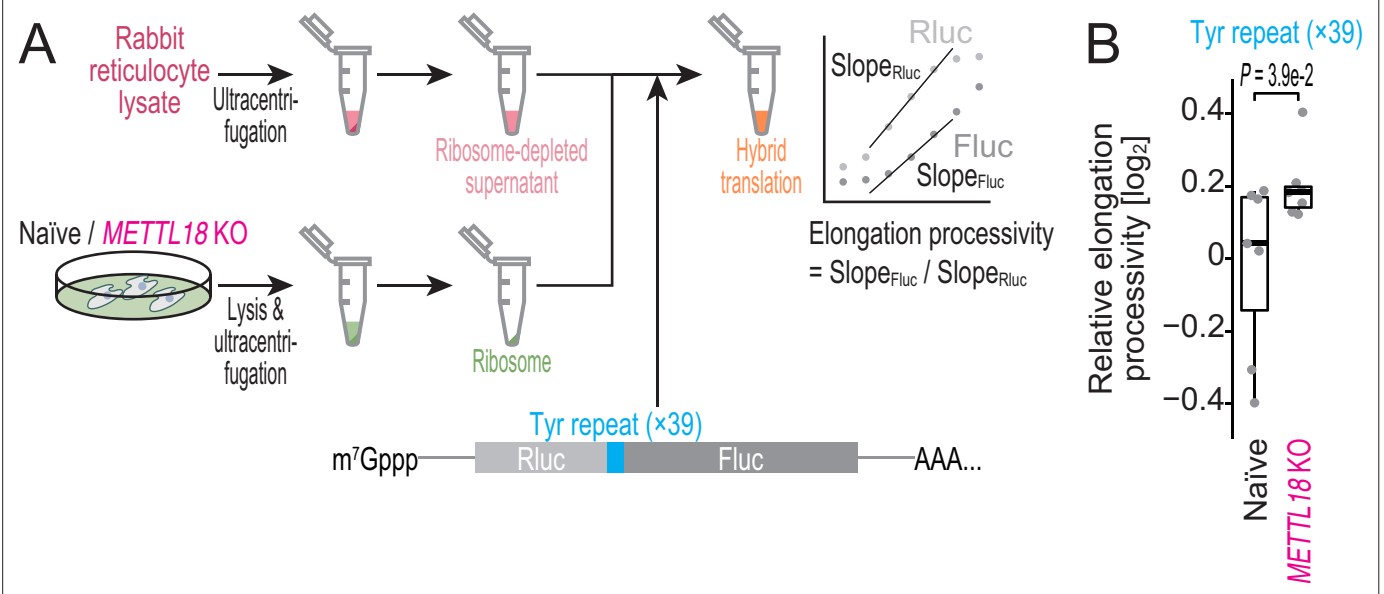

**Figure 5.** Ribosome without RPL3 methylation shows higher processivity on Tyr codons in vitro. (**A**) Schematic representation of the hybrid translation system and the processivity reporter. (**B**) The box plot for the relative ratio of Slope$_{Fluc}$ to Slope$_{Rluc}$ for the reporter with Tyr repeat insertion. Data from seven replicates (points) are shown. Significance was determined by Brunner–Munzel test (unpaired, two-sided).

The online version of this article includes the following source data and figure supplement(s) for figure 5:

**Source data 1.** Primary data for graphs in *Figure 5B*.

**Figure supplement 1.** Characterization of hybrid translation system and *Renilla*-firefly fused reporter.

**Figure supplement 1—source data 1.** Primary data for graphs in *Figure 5—figure supplement 1A, B*.

effects on Tyr codons, we used a repeat of the codon (×39; the number was due to cloning constraints). We note that the insertion of Tyr codon repeats reduced the elongation rate (or processivity), as we observed the reduced slope of the downstream Fluc synthesis (*Figure 5—figure supplement 1B*).

Then, we coupled the in vitro translation system and the reporter. We observed that RPL3 methylation-deficient ribosomes exhibited faster elongation on Tyr repeats than ones from naïve cells (*Figure 5B*).

These in cell and in vitro data together indicated that RPL3 methylation mediated a slowing of translation elongation on the Tyr codons.

## RPL3 histidine methylation ensures the proper proteostasis

Ribosome traverse along the mRNA determines the quality of protein synthesized (*Cassaignau et al., 2020*; *Collart and Weiss, 2019*; *Stein and Frydman, 2019*). Slowdown of ribosome elongation is advantageous since it allows the duration of nascent protein folding before completion of protein synthesis. Therefore, we reasoned that translation elongation slowdown at Tyr codons by RPL3 methylation facilitates protein folding on ribosomes and maintains proper homeostasis of the proteome.

To test this possibility, we employed an aggregation-prone firefly luciferase (Fluc) reporter (Arg188Gln-Arg261Gln double mutant or DM) fused to enhanced green fluorescent protein (EGFP) (*Gupta et al., 2011*). Since this engineered protein necessitates chaperones to fold properly, the reduced pool of available chaperones (i.e., proteotoxicity) leads to aggregation puncta of the reporter protein in cells. Indeed, METTL18 depletion induced the aggregation of FlucDM but not Fluc WT reporter (*Figure 6A and B*). As shown by the recovery of protein abundance by treatment with the proteasome inhibitor MG132 (*Figure 6C*), FlucDM protein was synthesized in *METTL18* KO cells of low quality and subjected to degradation.

Then, we explored the proteome, the quality of which was assisted by RPL3 histidine methylation. For this purpose, we surveyed the proteins aggregated in *METTL18* KO cells by SILAC (*Figure 7A*). We observed that a subset of proteins were enriched in precipitates of *METTL18* KO cell lysates (*Figure 7B*). This subgroup significantly accumulated Tyr-rich proteins (defined as proteins possessing

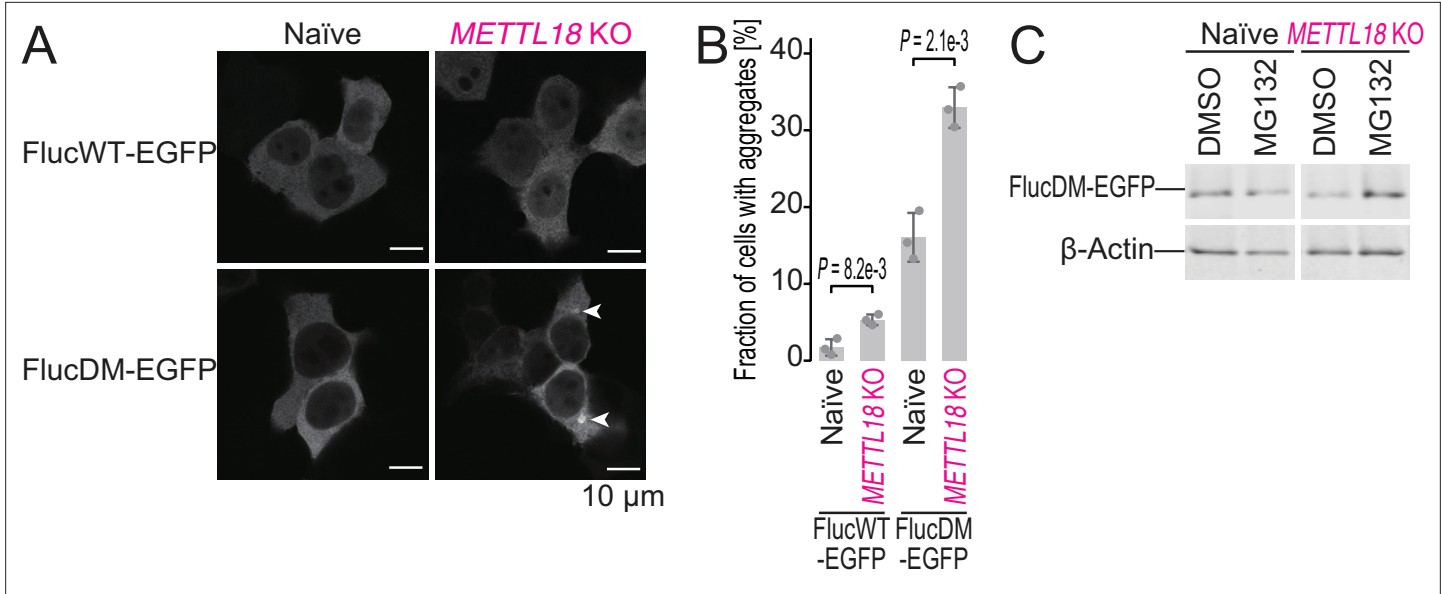

**Figure 6.** METTL18 deletion leads to cellular proteotoxicity. (**A**) Microscopic images of FlucWT-EGFP or FlucDM-EGFP in naïve and *METTL18* knockout (KO) HEK293T cells. Arrowhead, protein aggregation; scale bar, 10 µm. (**B**) Quantification of cells with Fluc-EGFP aggregates. Data from three replicates (points) and the mean (bar) with SD (error bar) are shown. Significance was determined by Student's *t*-test (unpaired, two-sided). (**C**) Western blot for FlucDM-EGFP (probed by anti-GFP antibody) expressed in naïve and *METTL18* KO HEK293T cells treated with MG132 (0.25 µM for 24 hr). β-Actin was probed as a loading control.

The online version of this article includes the following source data for figure 6:

**Source data 1.** Full and unedited images corresponding to *Figure 6A*.

**Source data 2.** Full and unedited blots corresponding to *Figure 6C*.

**Source data 3.** Primary data for graphs in *Figure 6B*.

30 Tyr or more) (hypergeometric test, p=0.0068) (*Figure 7B*). The high probability of protein precipitates could not be explained by the increased net protein synthesis measured by ribosome profiling (*Figure 7—figure supplement 1A*). More generally, Tyr-rich proteins were more prone to precipitate upon the deletion of METTL18 (*Figure 7—figure supplement 1B*) than proteins rich in other amino acids (*Figure 7C*). Thus, proteomic analysis of cellular precipitates revealed that the modulation of Tyr-specific translation elongation by RPL3 methylation confers proteome integrity.

These data led us to further investigate the properties of the proteome associated with Tyr. Here, we surveyed the aggregation propensity of the precipitated proteins by TANGO, which is based on statistical mechanics (*Fernandez-Escamilla et al., 2004*). Strikingly, metagene analysis showed prominent enrichment of the TANGO-predicted aggregation propensity around the Tyr codons (*Figure 7D*). As exemplified in the MACROH2A1 protein, the subpart of the aggregation-prone region in this protein was found on Tyr with reduced ribosome occupancies in *METTL18* KO cells (*Figure 7E*, *Figure 7—figure supplement 1C and D*).

Ultimately, the low-quality protein synthesized by RPL3-unmethylated ribosomes may serve as a substrate for proteasomes. To assess the fraction of proteins degraded by proteasome, we conducted SILAC analysis of the total proteome in *METTL18* KO cells treated with MG132, a proteasome inhibitor (*Figure 8A*). Indeed, this approach revealed a subset of more efficiently degraded proteins in *METTL18* KO cells (*Figure 8B*); this group of proteins tended to be reduced in *METTL18* KO cells and recovered by MG132 treatment. Moreover, this subset of proteins included Tyr-rich proteins (*Figure 8C*), which was congruent with the enhanced elongation observed in ribosome profiling.

The relationships among the aggregation-prone character of the protein, the cellular protein precipitates, increased proteasomal substrates, and reduced ribosome occupancy indicates that Tyr translation modulated by RPL3-methylated ribosomes is associated with the quality of the synthesized proteins, preventing unwanted protein aggregation (*Figure 9*).

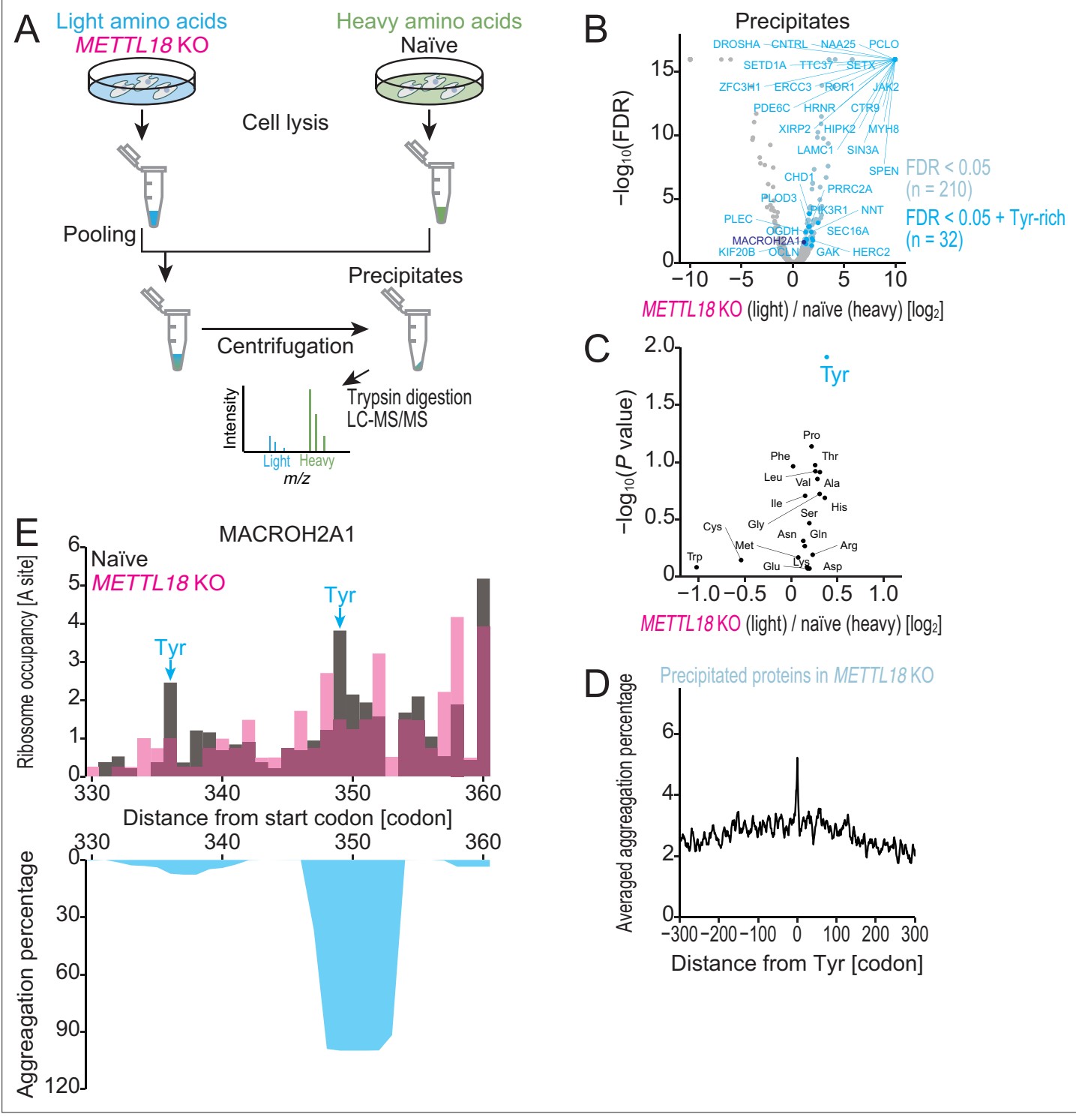

**Figure 7.** METTL18 deletion aggregates Tyr-rich proteins. (**A**) Schematic representation of SILAC-MS for precipitated proteins. (**B**) Volcano plot for precipitated proteins in *METTL18* knockout (KO) cells, assessed by SILAC-MS (n = 2). Tyr-rich proteins were defined as proteins with 30 or more Tyr residues. (**C**) Amino acids associated with protein precipitation in *METTL18* KO cells. Precipitated proteins enriched with each amino acid were compared to the total precipitated proteome. The mean fold change and the significance (Mann–Whitney *U*-test) were plotted. (**D**) Metagene plot for aggregation percentage, calculated with TANGO (*Fernandez-Escamilla et al., 2004*), around Tyr codons of precipitated proteins in *METTL18* KO cells (defined in **B**). (**E**) Distribution (at the A-site) of ribosome footprint occupancy (the mean of two independent experiments) along the MACROH2A1 gene in naïve (gray) and *METTL18* KO (magenta) HEK293T cells, depicted with the aggregation percentage (light blue) calculated by TANGO (*Fernandez-Escamilla et al., 2004*). Tyr codon positions are highlighted with arrows.

*Figure 7 continued on next page*

Figure 7 continued

The online version of this article includes the following figure supplement(s) for figure 7:

**Figure supplement 1.** Characterization of precipitated proteins identified by SILAC-MS.

## Discussion

In addition to SETD3 (*Dai et al., 2019*; *Guo et al., 2019*; *Kwiatkowski et al., 2018*; *Wilkinson et al., 2019*; *Zheng et al., 2020*), METTL18 provides a second example of $\tau$-*N*-methyltransferase. Whereas $\tau$-*N*-methylation at the equivalent residue (His243) in yeast Rpl3 has been indirectly demonstrated (*Webb et al., 2010*), this study provided solid evidence (such as MS and cryo-EM) of the modification on the homolog in humans.

Similar to prior yeast work (*Al-Hadid et al., 2014*), human RPL3 alone could not be methylated by METTL18. Rather, RPL3 was most likely to be modified in the early 60S biogenesis intermediate complex. This suggests that METTL18 recognizes a unique interface formed by RPL3 and other ribosome proteins and/or assembly factors. In the recent structural studies on pre-60 assembly intermediates (*Kater et al., 2017*; *Sanghai et al., 2018*), His245 of RPL3 in the early pre-60S in state B (*Kater et al., 2017*) is exposed to solvent (*Figure 9—figure supplement 1A*), possibly enabling access to METTL18, although His245 per se was not visible in the structure because of flexibility. In contrast, ribosomal proteins and rRNAs in the later stage of 60S biogenesis (at state D) (*Kater et al., 2017*) fill the corresponding space (*Figure 9—figure supplement 1B*). Thus, histidine methylation by METTL18 should be restricted to specific timing of the macromolecule assembly.

In the course of this work, a paper from Falnes and coworkers was published and reached a similar conclusion regarding the methylation site (His245 in RPL3) and the methylation type ($\tau$-*N* position) mediated by human METTL18 (*Małecki et al., 2021*). Although the earlier report showed that recombinant METTL18 protein methylated isolated ribosomes from *METTL18* KO, the authors also observed that METTL18 localized in nucleoli (*Małecki et al., 2021*) where ribosomes are still under assembly. These data could be interpreted to show that the purified ribosome may consist of contaminated pre-60S, and this fraction was an efficient substrate of METTL18.

The impact of METTL18 loss on translation found in this study differed from that in the earlier report (*Małecki et al., 2021*). In the earlier report, ribosome profiling in *METTL18* KO HAP1 cells revealed widespread effects of translation elongation, exemplified by slowdown of the GAA codon (*Małecki et al., 2021*). In contrast, the same experiment in this work with HEK293T and HAP1 (including the same cell line [2-nt del.] used in *Małecki et al., 2021*) showed that *METTL18* KO enhanced ribosome traversal on Tyr codons (*Figure 4*). We did not have any information on how to reconcile this inconsistency in ribosome profiling data. At least, as reported previously (*Małecki et al., 2021*), we have made a similar observation in the bulk ribosomal complex abundance in the *METTL18* KO HAP1 cell line (2-nt del.); reduced polysomes (*Figure 9—figure supplement 2A and B*) and unaltered 40S and 60S (*Figure 9—figure supplement 2C–E*). The effect of RPL3 methylation on translation elongation could be dynamic and dependent on cellular status affected by circumstances (e.g., culturing conditions). Future studies across diverse tissues and cellar status will provide a global look at the condition-dependent effects of the ribosome protein methylation.

The modification-mediated ribosome slowdown revealed in this study provides a unique example of translational control since modification in general often offers a smooth translation, as exemplified in tRNA modifications (*Nedialkova and Leidel, 2015*; *Tuorto et al., 2018*). The exact mechanism of how His245 methylation in RPL3 leads to Tyr-specific elongation retardation remains elusive. In bacteria, the loop of helix 35, where unmethylated His245 interacts, was heavily modified (*Kannan and Mankin, 2011*), although the role of rRNA modification has been unclear. Eukaryotic RPL3 may functionally replace RNA modification by histidine methylation. From another viewpoint, His245 is located in the 'basic thumb' region (Arg234-Arg246), which bridges the interaction of 28S rRNA helices (H61, H73, and H90) and facilitates the formation of a so-called 'aminoacyl-tRNA accommodation corridor' (*Meskauskas and Dinman, 2010*). Thus, the methylation of His245 may allosterically alter the dynamic character of the aminoacyl-tRNA accommodation corridor to restrict the movement of charged Tyr on tRNA. Indeed, conformational changes in the aminoacyl-tRNA accommodation corridor were suggested by molecular dynamics simulation (*Gulay et al., 2017*).

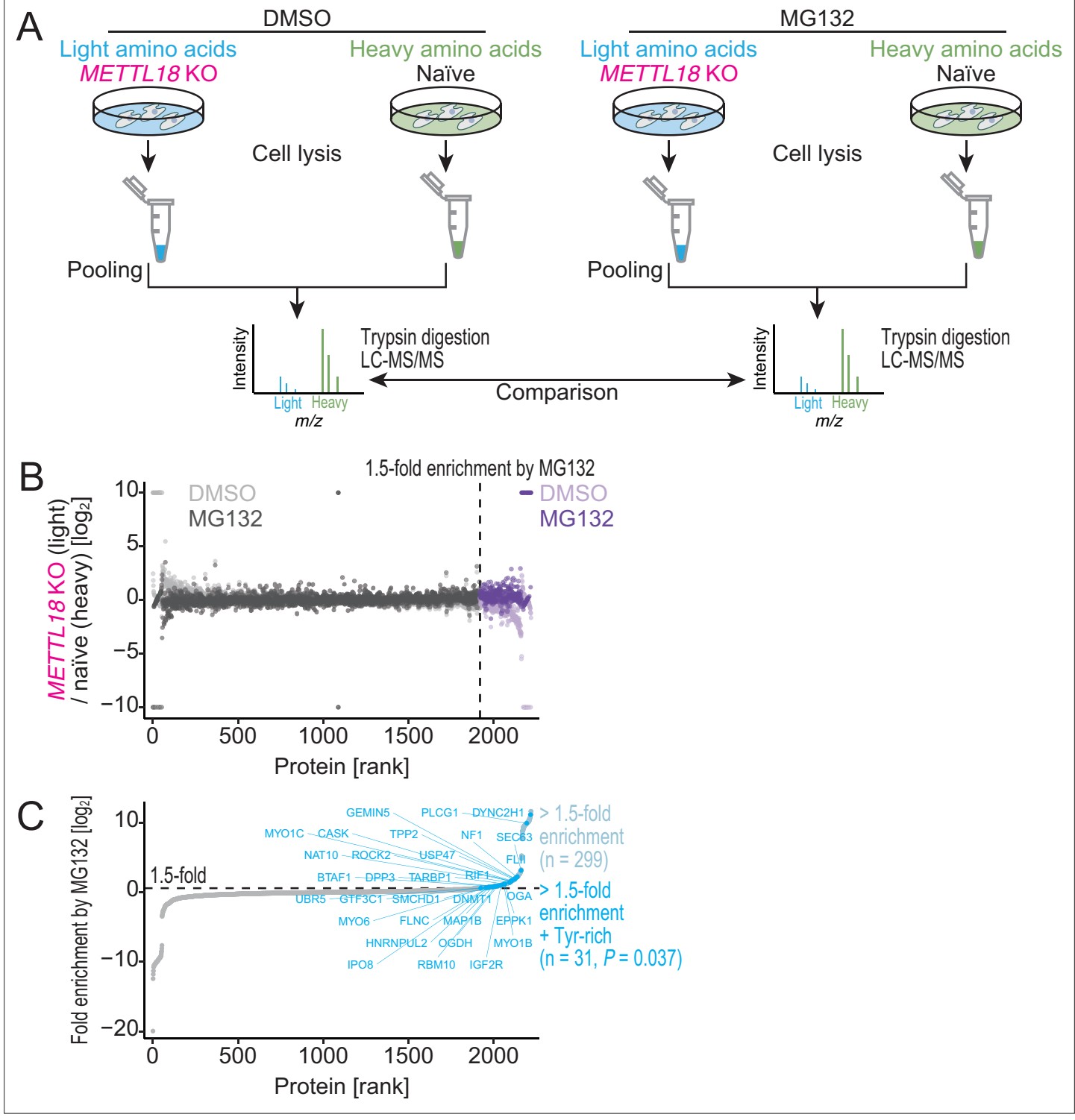

**Figure 8.** METTL18 deletion degrades Tyr-rich proteins by proteasome. (**A**) Schematic representation of SILAC-MS for total proteins. (**B, C**) Cellular protein abundance changes in *METTL18* knockout (KO) cells with the treatment of proteasome inhibitor MG132 and the control DMSO, assessed by SILAC-MS (n = 2). The relative abundance in *METTL18* KO HEK293T cells compared to naïve HEK293T cells is calculated. Data with DMSO or MG132 treatment (**B**) and fold enrichment (MG132 compared to DMSO) (**C**) are shown ranked by the fold enrichment in (**C**). Proteins with 1.5 or higher enrichment by MG132 treatment are highlighted. Tyr-rich proteins are defined as proteins with 30 or more Tyr residues.

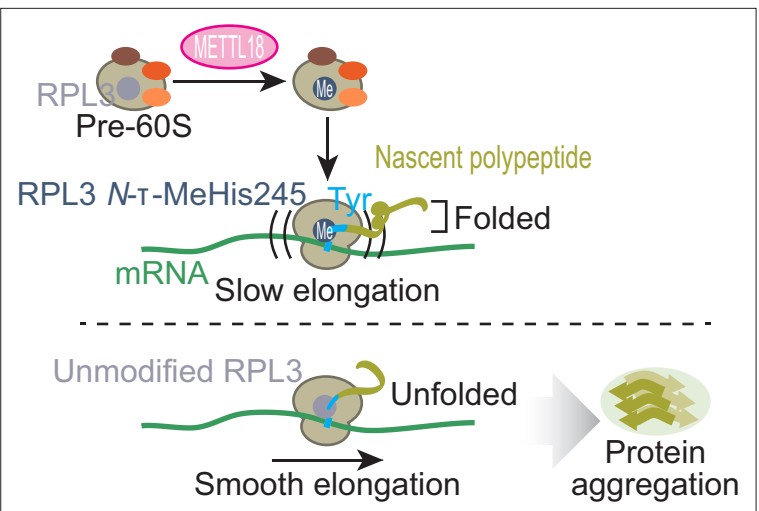

**Figure 9.** Schematic representation of METTL18-mediated control of translation and proteostasis. METTL18 adds a methyl moiety at the $\tau$-$N$ position of His245 in RPL3 in the form of an early 60S biogenesis intermediate. Methylated ribosomes slow elongation at Tyr codons and extend the duration of nascent peptide folding, ensuring proteostatic integrity. Without RPL3 methylation, the accumulation of unfolded and ultimately aggregated proteins in cells was induced.

The online version of this article includes the following source data and figure supplement(s) for figure 9:

**Figure supplement 1.** Comparison of the structure of the early and late pre-60S as a possible RPL3 methylation target.

**Figure supplement 2.** The impacts of METTL18 deletion in HAP1 cells on ribosomal complex formation.

**Figure supplement 2—source data 1.** Primary data for graphs in *Figure 9—figure supplement 2B, D, and E*.

We also did not exclude the possibility that this modification may impact other processes of translation. Indeed, an earlier report in yeast suggested that Hpm1 (a homolog of METTL18) deletion decreased the fidelity of translation, inducing stop codon readthrough (*Al-Hadid et al., 2016*; *Al-Hadid et al., 2014*). However, our ribosome profiling data in HEK293T cells did not show any increase in ribosome footprints in the 3' UTR (*Figure 9—figure supplement 1C and D*) that could be accumulated by the stop codon readthrough (*Arribere et al., 2016*; *Dunn et al., 2013*).

Although ribosome traversal along the CDS is generally defined by the decoding rate in bacteria (*Mohammad et al., 2019*) and yeasts (*Hussmann et al., 2015*; *Wu et al., 2019*), the codon/amino acid-specific effects of RPL3 histidine methylation in humans suggest that the landscape is more complex in higher eukaryotes. Indeed, ribosome occupancy in mammals is poorly predicted by tRNA abundance in cells (*Han et al., 2020*). Thus, it is not surprising that a wide array of ribosome modifications, including RPL3 histidine modification, ultimately define the speed of ribosome movement and thus the quality of protein synthesized in cells.

## Materials and methods
### Plasmid construction
#### PX330-B/B-gMETTL18
To express two guide RNAs targeting upstream and downstream regions of exon 2 of the *METTL18* gene, DNA fragments containing 5'-TCTCTTTAGCAGCTTATACA-3' and 5'-GGTTGTGGATCAGGTTT ACT-3' were cloned into PX330-B/B (*Yamazaki et al., 2018*) via the BbsI and BsaI sites, respectively.

#### pL-CRISPR.EFS.tRFP-gSETD3
To express a guide RNA targeting exon 6 of the *SETD3* gene, a DNA fragment containing 5'-AGCCA TGGGAAACATCGCAC-3' was cloned into pL-CRISPR.EFS.tRFP (Addgene#57819; http://n2t.net/ addgene:57819; RRID:Addgene_57819) (*Heckl et al., 2014*).

### pET19b-mMETTL18

To express N-terminally His-tagged full-length mouse METTL18, cDNA was PCR-amplified from a FANTOM clone (AK139786) and cloned into NdeI and XhoI sites of the pET19b vector (Novagen).

### pCold-GST-mMETTL18

To express N-terminally His- and GST-tagged full-length mouse METTL18, the PCR-amplified fragment was cloned into the NdeI and XhoI sites of the pCold-GST vector (TaKaRa).

### pcDNA3-hRPL3-FLAG (WT and His245Ala) and pcDNA3-mRPL3-FLAG (WT and His245Ala)

To express C-terminally FLAG-tagged mouse and human RPL3, each cDNA fragment was amplified from the mouse cDNA and HEK293T cDNA libraries and cloned into EcoRI and NotI sites of the pcDNA3 vector (Invitrogen) with a C-terminal FLAG-tag sequence. To generate His245Ala mutants, the QuikChange Site-Directed Mutagenesis Kit (Agilent Technologies) was used.

### pQCXIP-hMETTL18-HA and hMETTL18-Asp193Lys-Gly195Arg-Gly197Arg-HA

For retrovirus expression, cDNA fragment of human METTL18 with a C-terminal HA sequence was cloned into AgeI and EcoRI sites of the pQCXIP vector (Clontech). To generate hMETTL18-Asp193Lys-Gly195Arg-Gly197Arg-HA, the QuikChange Site-Directed Mutagenesis Kit (Agilent Technologies) was used.

### psiCHECK2-Y0× and Y39×

To generate the psiCHECK2-Y0× reporter, the intergenic region between Rluc and Fluc was excluded by inverse PCR. The plasmid encodes the ORF for the Rluc-Fluc fusion. To generate the psiCHECK2-Y39× reporter, 39 repeats of the TAC sequence were inserted between Rluc and Fluc of psiCHECK2-Y0×.

## Cell lines: HEK293T

HEK293T KO cell lines were generated by the CRISPR-Cas9 system. All cell lines in the laboratory were routinely tested for *Mycoplasma* contamination with the MycoAlert Mycoplasma Detection Kit (LONZA) and confirmed negative.

### *METTL18* KO cells

PX330-B/B-gMETTL18, which expresses hCas9 and two guide RNAs designed to induce large deletions in exon 2, and pEGFP-C1 (Clontech) were cotransfected into HEK293T cells. After 2 days of incubation, individual GFP-positive cells were sorted into 96-well plates. The clonal cell lines were screened by genomic PCR with the primers 5'-GGACTTTATGTTTGTCCAGGTGG-3' and 5'-TGGGTTGTAAATG GTTTCTGAGG-3'. e-Myco VALiD (LiliF) confirmed that the cells were negative for *Mycoplasma*.

### *METTL18* KO cells with stable METTL18 expression

Retrovirus packaging cells were transfected with pQCXIP-hMETTL18-HA, pQCXIP-hMETTL18-Asp193Lys-Gly195Arg-Gly197Arg-HA, or the pQCXIP empty vector (as a negative control) using PEI transfection reagent (Polysciences) and cultured for 24 hr. Then, *METTL18* KO cells were inoculated with the virus-containing culture supernatant and 4 µg/ml of polybrene. At 24 hr post-infection, cells were selected with 1 µg/ml puromycin and cultured for additional 2 weeks. e-Myco VALiD (LiliF) confirmed that the cells were negative for *Mycoplasma*.

### SETD3 KO and SETD3-METTL18 DKO cells

pL-CRISPR.EFS.tRFP-gSETD3, which expressed a guide RNA designed to induce small insertion or deletion (InDel), was transfected into HEK293T cells or *METTL18* KO cells. After 2 days of incubation, individual RFP-positive cells were sorted into 96-well plates. The clonal cell lines were further screened by Western blot for SETD3 protein.

## Cell lines: HAP1

HAP1 cells, including naïve cells (C631) and *METTL18* KO cells (1-nt del., HZGHC000541c009; 2-nt del., HZGHC000541c002; and 4-nt del., HZGHC000541c012), were purchased from Horizon Discovery. e-Myco VALiD (LiliF) confirmed that the cells were negative for *Mycoplasma*.

## Recombinant protein purification

### *Salmonella* MTAN, His-METTL18, and His-GST-METTL18

The BL21 (pLysS) strain transformed with *Salmonella* MTAN (a gift from Vern Schramm [Addgene plasmid #64041; http://n2t.net/addgene:64041; RRID:Addgene_64041]), pET19b-mMETTL18, or pCold-GST-mMETTL18 was cultured in 2× YT medium with 100 µg/ml ampicillin and 0.2 mM isopropyl β-D-1-thiogalactopyranoside (IPTG) for 18 hr at 16°C. The pelleted cells were lysed with 1× PBS with 0.5% NP-40 by sonication with a sonic homogenizer (Branson Ultrasonics, Sonifier S-250D) for 5 min on ice. After centrifugation at 15,000 × *g* for 10 min, the cleared cell extract was incubated with Ni-NTA Agarose (QIAGEN) or Glutathione Sepharose 4B (Cytiva) for 1 hr at 4°C with gentle agitation. The beads were washed five times with His wash buffer (50 mM Tris-HCl pH 7.4 and 25 mM imidazole) or GST wash buffer (1× phosphate-buffered saline [PBS]). Then, proteins were eluted with His elution buffer (50 mM Tris-HCl pH 7.4 and 250 mM imidazole) or GST elution buffer (50 mM Tris-HCl pH 8.0 and 50 mM glutathione). The purified proteins were dialyzed with dialysis buffer (50 mM Tris-HCl pH 8.0, 100 mM NaCl, 0.2 mM dithiothreitol [DTT], and 10% glycerol) using Slide-A-Lyzer Dialysis Cassettes (MWCO 10 kDa, Thermo Fisher Scientific) or Amicon Ultra (MWCO 10 kDa, Merck Millipore). The protein concentration was measured using the Bradford Protein Assay Kit (Bio-Rad).

## Western blot

Anti-α-tubulin (Sigma-Aldrich, clone B-5-1-2, 1:1000), anti-METTL18 (Proteintech Group, 25553-1-AP), anti-SETD3 (Abcam, ab174662, 1:1000), anti-RPL3 (Proteintech Group, 66130-1-lg and 11005-1-AP, 1:1000), anti-PES1 (Abcam, ab252849, 1:1000), anti-NMD3 (Abcam, ab170898, 1:1000), anti-HA (Medical & Biological Laboratories [MBL], M180-3, 1:1000), anti-GFP (Abcam, ab6556, 1:1000), anti-RPL17 (Proteintech Group, 14121–1-AP, 1:1000), and anti-β-actin (MBL, M177-3, 1:1000) primary antibodies were used.

For *Figures 1C and 2A*, *Figure 1—figure supplement 1D*, anti-mouse IgG, HRP-Linked Whole Ab Sheep (Cytiva, NA931V, 1:5000) and anti-rabbit IgG, HRP-Linked Whole Ab Donkey (Cytiva, NA934V, 1:5000) secondary antibodies were used. The chemiluminescence was raised with a Western Lightning Plus-ECL Kit (PerkinElmer) according to the manufacturer's protocol and detected with X-ray film (FUJIFILM, RX-U). For biotinylated proteins, high-sensitivity streptavidin-HRP (Thermo Fisher Scientific, 21130) was used.

To generate *Figures 2B and 5C*, *Figure 3—figure supplement 1D*, IRDye680- or IRDye800CW-conjugated secondary antibodies (LI-COR Biosciences, 925-68070/71 and 926-32210/11/19, respectively, 1:10,000) were used. Images were obtained with Odyssey CLx (LI-COR Biosciences).

## Mass spectrometry

### Multiple reaction monitoring

Methylhistidine content analysis was performed essentially as previously described (*Davydova et al., 2021*). Proteins were precipitated with acetone and hydrolyzed to amino acids with 6 N HCl at 110°C for 24 hr. After dissolving in 25 µl of 5 mM ammonium formate/0.001% formic acid, the amino acids were applied to a LC system (Thermo Fisher Scientific, Vanquish UHPLC). The amino acids loaded on a C18 column (YMC, YMC-Triart C18, 2.0 × 100 mm length, 1.9 µm particle size) were separated at a flow rate of 0.3 ml/min by gradient elution of mobile phase 'A' (5 mM ammonium formate with 0.001% formic acid) and mobile phase 'B' (acetonitrile) as follows: 0/0 – 1.5/0 – 2/95 – 4/95 – 4.1/0 – 7/0 (min/%B). The effluent was then directed to an electrospray ion source (Thermo Fisher Scientific, HESI-II) connected to a triple quadrupole mass spectrometer (Thermo Fisher Scientific, TSQ Vantage EMR) in positive ion MRM mode. The electrospray was run with the following settings: spray voltage of 3000 V, vaporizer temperature of 450°C, sheath gas pressure of 50 arbitrary units, auxiliary gas pressure of 15 arbitrary units, and collision gas pressure of 1.0 mTorr. The transition of specific MH+→ fragment ions was monitored (His, *m/z* 156.1→83.3, 93.2, and 110.2; π-*N*-MeHis, *m/z* 170.1→95.3, 97.3, and 109.2; τ-*N*-MeHis, *m/z* 170.1→81.3, 83.3, and 124.2). Data were calibrated with 1–250 nM

standards (His, π-*N*-MeHis, and τ-*N*-MeHis) at every run. The concentrations of His, π-*N*-MeHis, and τ-*N*-MeHis in the samples were calculated from the calibration curves obtained from the standards.

## ProSeAM-SILAC-MS

ProSeAM was synthesized as previously described (*Sohtome et al., 2018*). ProSeAM substrate screening was carried out as reported (*Shimazu et al., 2018*) with some modifications. *METTL18* KO cells were cultured in Dulbecco's modified Eagle's medium (DMEM) containing either light Arg/Lys or heavy isotope-labeled Arg ($^{13}C_6$ $^{15}N_4$ L-arginine)/Lys ($^{13}C_6$ $^{15}N_2$ L-lysine) (Thermo Fisher Scientific, respectively) at least six doubling times. The cells were lysed with 50 mM Tris-HCl pH 8.0, 50 mM KCl, 10% glycerol, and 1% *n*-dodecyl-β-ᴅ-maltoside. For the cells cultured with the heavy amino acids, the lysate containing 200 µg of proteins was incubated with 150 µM ProSeAM and 10 µg of His-METTL18 in 50 mM Tris-HCl pH 8.0 at 20°C for 2 hr. For the lysate with the light amino acids, His-METTL18 was omitted from the reaction. The reaction was stopped by the addition of four volumes of ice-cold acetone. The proteins were precipitated by centrifugation at 15,000 × *g* for 5 min, washed once with ice-cold acetone, and then dissolved in 58.5 µl of 1× PBS containing 0.2% sodium dodecyl sulfate (SDS). After the addition of 15 µl of 5× click reaction buffer (7.5 mM sodium ascorbate [Nacalai Tesque], 0.5 mM TBTA [AnaSpec], and 5 mM $CuSO_4$) and 1.5 µl of 10 mM Azide-PEG4-Biotin (Click Chemistry Tools), the click reaction was conducted for 60 min at room temperature, stopped with four volumes of ice-cold acetone, and precipitated as described above. The protein in the pellet was resuspended in 75 µl of binding buffer (1× PBS, 0.1% Tween-20, 2% SDS, and 20 mM DTT) and sonicated for 10 s. The light and heavy isotope-labeled samples were pooled in 450 µl of IP buffer (Tris-buffered saline [TBS] and 0.1% Tween-20) containing 3 µg of Dynabeads M-280 Streptavidin (Thermo Fisher Scientific) and incubated for 30 min at room temperature (note that the final SDS concentration in the solution was 0.5%). The protein-bound beads were washed three times with bead wash buffer (1× PBS, 0.1% Tween-20, and 0.5% SDS) and twice with 100 mM ammonium bicarbonate (ABC) and then used for Western blotting and MS.

For MS/MS analysis, the beads were incubated in 20 mM DTT and 100 mM ABC for 30 min at 56°C and then for 30 min at 37°C in the dark with supplementation with 30 mM iodoacetamide. Subsequently, proteins were digested with 1 µg trypsin (Promega) and subjected to LC (Thermo Fisher Scientific, EASY-nLC 1000) coupled to a Q Exactive Hybrid Quadrupole-Orbitrap Mass Spectrometer (Thermo Fisher Scientific) with a nanospray ion source in positive mode, as previously reported (*Davydova et al., 2021*). The peptides loaded on a NANO-HPLC C18 capillary column (0.075 mm inner diameter × 150 mm length, 3 µm particle size, Nikkyo Technos) were eluted at a flow rate of 300 nl/min with the two different slopes of mobile phase 'A' (water with 0.1% formic acid) and mobile phase 'B' (acetonitrile with 0.1% formic acid): 0–30% of phase B in 100 min and 30–65% of phase B in 20 min. Subsequently, the mass spectrometer in the top 10 data-dependent scan mode was run with the following parameters: spray voltage, 2.3 kV; capillary temperature, 275°C; mass-to-charge ratio, 350–1800; normalized collision energy, 28%. The MS and MS/MS data obtained with Xcalibur software (Thermo Fisher Scientific) were surveyed in the Swiss-Prot database with Proteome Discoverer (version 2.3, Thermo Fisher Scientific) and MASCOT search engine software (version 2.7, Matrix Science). Peptides with false discovery rates (FDRs) less than 1% were considered. Proteins with the following criteria were defined as METTL18-dependent labeled proteins: 1.5-fold or more increase in heavy amino acid sample compared to light amino acid sample; one, 10% or more coverage of the protein; and three or more peptides identified.

## LC-MS/MS for methylated peptide

The SDS-PAGE-separated and Coomassie staining-visualized proteins were excised and destained. The gel slices were reduced with 50 mM DTT and 4 M guanidine-HCl at 37°C for 2 hr, followed by alkylation with 100 mM acrylamide at 25°C for 30 min. The RPL3 proteins were digested with chymotrypsin. The MS and MS/MS spectra were acquired with a Q Exactive HFX (Thermo Fisher Scientific). The mass spectrometer was operated in positive mode. The MS/MS spectra were obtained using a data-dependent top 10 method. The acquired data were processed using Proteome Discoverer (version 2.3, Thermo Fisher Scientific). The processed data were used to search with MASCOT (version 2.7, Matrix Science) against the in-house database including the amino acid sequences of RPL3 using the following parameters: type of search, MS/MS ion search; enzyme, none; fixed modification, none;

variable modifications, Gln->pyro-Glu (N-term Q), oxidation (M), oropionamide (C), and methyl (H); mass values, monoisotopic; peptide mass tolerance, ±15 ppm; fragment mass tolerance, ±30 mmu; peptide charge, 1+, 2+, and 3+; instrument type, ESI-TRAP.

## SILAC-MS for *METTL18* KO

Isotopically heavy amino acids (0.1 mg/ml $^{13}C_6^{15}N_2$ L-lysine-HCl and 0.1 mg/ml $^{13}C_6^{15}N_4$ L-arginine-HCl [both FUJIFILM Wako Chemicals]) or regular amino acids (0.1 mg/ml L-lysine-HCl and 0.1 mg/ml L-arginine-HCl [both FUJIFILM Wako Chemicals]) were added to DMEM deficient in both L-lysine and L-arginine for SILAC (Thermo Fisher Scientific) supplemented with 10% dialyzed fetal bovine serum (FBS) (Sigma-Aldrich). Naïve HEK293T cells and *METTL18* KO cells were cultured in media with heavy and light isotopes, respectively, for 2 weeks.

After a brief wash with PBS, cells were lysed with buffer containing 20 mM Tris-Cl pH 7.5, 150 mM KCl, 5 mM MgCl₂, 1% Triton X-100, and 1 mM DTT. The protein concentration was measured using Qubit 2.0 Fluorometer (Thermo Fisher Scientific). The same amounts of proteins from light and heavy isotope-labeled lysates were mixed and centrifuged at 500 × *g* for 3 min. The supernatant was further centrifuged at 20,000 × *g* and 4°C for 15 min. The precipitate was collected and used for analysis.

Filter-aided sample preparation (FASP) was used for protein digestion (*Wiśniewski et al., 2009*). The precipitate was dissolved in SDT lysis buffer (4% [w/v] SDS, 0.1 M DTT, and 100 mM Tris-HCl pH 7.6) and incubated at 95°C for 5 min. Denatured samples were diluted 10 times with UA buffer (8 M urea and 100 mM Tris-HCl pH 8.5) and loaded to Vivacon 500, 30000 MWCO Hydrosart (Sartorius) to trap the protein on the filter. The filter unit was washed with the UA buffer. Proteins on the filter unit were alkylated with 100 μl of the IAA solution (0.05 M iodoacetamide in UA). Then, the filter unit was washed with 100 μl of UA buffer three times and 100 μl of 50 mM NH₄HCO₃ three times. Proteins on the filter unit were digested with 40 μl of trypsin solution (trypsin [V5111, Promega] in 50 mM NH₄HCO₃) at 37°C for 18 hr. Then, the peptides were eluted from the filter unit by 40 μl of NH₄HCO₃ twice and 50 μl of 0.5 M NaCl once and then pooled.

LC-MS/MS analysis was performed using EASY-nLC 1000 (Thermo Fisher Scientific) and Q Exactive (Thermo Fisher Scientific) equipped with a nanospray ion source. The peptides were separated with a NANO-HPLC capillary column C18 (0.075 × 150 mm, 3 μm, Nikkyo Technos) at 300 nl/min flow rate with strep gradients of solvent A (0.1% formic acid) and solvent B (acetonitrile with 0.1% formic acid); 0–30% B for 100 min and then 30–65% B for 20 min. The resulting MS and MS/MS data were searched against the Swiss-Prot database using Proteome Discoverer (version 2.4, Thermo Fisher Scientific) with MASCOT search engine software (version 2.7, Matrix Science). The peptides with FDR values of 0.05 or less were considered for the subsequent analysis. To quantify the difference of peptides labeled with differential isotopes, the built-in SILAC 2-plex quantification method in Proteome Discoverer (version 2.4, Thermo Fisher Scientific) was used. Peptide abundances were normalized by total peptide amount. p-Values were calculated by the background-based *t*-test and then adjusted by the Benjamini–Hochberg method.

For total proteome analysis, we treated cells with 0.25 μM MG132 (FUJIFILM Wako Chemicals) for 24 hr before cell harvest. Cell lysates were prepared as described above, and the supernatant obtained by centrifugation at 20,000 × *g* and 4°C for 15 min was used for the downstream analysis.

## Methylation assay

FLAG-tagged RPL3 was transiently expressed in *METTL18* KO cells and immunopurified with ANTI-FLAG M2 Affinity Gel (Sigma-Aldrich). The purified proteins were incubated in 1× reaction buffer (50 mM Tris-HCl pH 8.5 and 50 mM MgCl₂) with 1 μg of His-GST-METTL18, 2 μM MTAN, and 0.01 μCi of $^{14}$C-labeled SAM (PerkinElmer) at 30°C for 2 hr. The reaction was stopped by the addition of Laemmli SDS-sample buffer. Proteins were separated on a 10% acrylamide SDS-PAGE gel. The dried gel was exposed to an imaging plate (FUJIFILM) for 48 hr. The autoradiograph was detected with a phosphor imaging scanner (Cytiva, Amersham Typhoon).

## Cryo-EM

The crude ribosomal pellet was suspended in buffer A (50 mM Tris-HCl pH 7.5, 150 mM KCl, 4 mM magnesium acetate, 1 mM DTT, 7% [w/v] sucrose, and 1 mM puromycin) by mixing with a magnetic stirrer for 4 hr on ice. After stirring, the suspension was centrifuged at 15,000 × *g* for 10 min at

4°C. The supernatant was loaded onto a HiPrep 16/60 Sephacryl S-500 column (Cytiva) equilibrated with buffer A without puromycin. The fraction was recovered and concentrated using Amicon Ultra (MWCO 50 kDa, Merck Millipore). The concentrated mixture was supplemented with 0.1 mM puromycin, 2 mM magnesium acetate, and 2 mM ATP and incubated for 30 min at 37°C. The reaction was loaded onto a 10–40% (w/v) sucrose gradient with buffer B (50 mM Tris-HCl pH 7.5, 500 mM KCl, 4 mM magnesium acetate, and 2 mM DTT) and centrifuged at 25,000 rpm in an SW28 rotor for 16 hr at 4°C. 2 ml fractions were successively fractionated from the top of the gradient. An aliquot of each fraction was analyzed by SDS-PAGE, and the gels were stained with Coomassie brilliant blue (CBB) to detect the 40S and 60S subunit proteins. The 40S- and 60S-subunit fractions were mixed and concentrated using Amicon Ultra filter units (MWCO 50 kDa, Merck Millipore). The concentrated mixture was loaded onto a 10–50% (w/v) sucrose gradient with buffer C (50 mM Tris-HCl pH 7.5, 150 mM KCl, 10 mM magnesium acetate, and 2 mM DTT) and centrifuged at 28,000 rpm in an SW41Ti rotor for 3 hr at 4°C. Gradient fractionation was carried out using a piston gradient fractionator equipped with a TRIAX flow cell detector (BioComp) by continuous monitoring of absorbance at a wavelength of 280 nm. The fractions containing 80S ribosomes were dialyzed against preparation buffer (10 mM HEPES-KOH pH 7.5, 30 mM potassium acetate, 10 mM magnesium acetate, and 1 mM DTT) and concentrated using Amicon Ultra filter units (MWCO 50 kDa, Merck Millipore). The concentrated sample was quantified by measuring absorbance at a wavelength of 260 nm and flash-cooled with liquid nitrogen.

For cryo-EM sample preparation, Quantifoil R1.2/1.3 300 mesh copper grids (Quantifoil) were covered with an amorphous carbon layer prepared in-house. The thawed sample was diluted to 70 nM (an absorbance at 260 nm of 3.5) with preparation buffer, and 3 µl of the sample was applied onto grids at 4°C at 100% relative humidity using a Vitrobot Mark IV (FEI). After incubation for 30 s and blotting for 3 s, the grids were plunged into liquid ethane.

The cryo-EM dataset was collected with a Tecnai Arctica transmission electron microscope (FEI) operated at 200 kV using a K2 summit direct electron detector (Gatan) (0.97 Å/pixel). The 5517 images collected were fractionated to 40 frames, with a total dose of ~50 e⁻/Å$^2$.

Processing of cryo-EM data was performed with RELION-3.1 (*Zivanov et al., 2020*). The movie frames were aligned with MotionCor2 (RELION's own implementation), and the CTF parameters were estimated with CTFFIND-4.1 (*Rohou and Grigorieff, 2015*). Particles were automatically picked using a template-free Laplacian-of-Gaussian (LoG) filter (250–500 Å), and 381,227 particles were extracted with twofold binning. After 2D classification, 136,688 particles were selected and applied to 3D classification. A low-pass-filtered (40 Å) map of the human 80S ribosome (EMD-9701) (*Yokoyama et al., 2019*) was used as a reference map for 3D classification, and 118,470 particles were selected after this step. These particles were re-extracted without rescaling and used in 3D refinement, Bayesian polishing, CTF refinement, and then 3D refinement again. After these steps, focused refinement with a mask on the 60S subunit and postprocessing resulted in a resolution of 2.72 Å.

For molecular modeling, the model of the 60S subunit from the human 80S ribosome structure at 2.9 Å resolution (PDB: 6QZP) (*Natchiar et al., 2017*) was used as a starting model and manually fitted into the map using UCSF Chimera (*Pettersen et al., 2004*). Map sharpening and model refinement were performed in PHENIX (*Adams et al., 2010*), and the model was further refined manually with Coot (*Emsley et al., 2010*). The modified nucleotides of ribosomal RNAs were introduced based on quantitative MS data (*Taoka et al., 2018*).

## Sucrose density gradient

For *Figure 2B*, cells were lysed with whole-cell lysis buffer (50 mM Tris-HCl pH 7.5, 150 mM NaCl, 5 mM MgCl$_2$, 1% NP-40, 1 mM DTT, and 100 µg/ml cycloheximide). The whole-cell lysates were passed through a 21-gauge needle and incubated at 4°C for 15 min. To obtain *Figure 4—figure supplement 1A–D* and *Figure 9—figure supplement 2A and B*, cells were lysed with lysis buffer (20 mM Tris-Cl pH 7.5, 150 mM KCl, 5 mM MgCl$_2$, 1% Triton X-100, 1 mM DTT, and 100 µg/ml cycloheximide) and cleared by centrifugation at 20,000 × *g* and 4°C for 10 min. Cell lysate containing 40 µg of total RNA was loaded onto a 10–50% sucrose gradient and ultracentrifuged at 35,300 rpm and 4°C for 2.5 hr by a Himac CP80WX ultracentrifuge (Hitachi) with a P40ST rotor (Hitachi).

To obtain *Figure 1—figure supplement 2A*, *Figure 3—figure supplement 1B, C, E, and F*, and *Figure 9—figure supplement 2C and D*, cells were lysed with EDTA lysis buffer (20 mM Tris-Cl pH 7.5,

150 mM KCl, 5 mM EDTA, 1% Triton X-100, and 1 mM DTT) and centrifuged at 20,000 × *g* and 4°C for 10 min. The supernatant with 40 μg of total RNA was ultracentrifuged at 35,300 rpm and 4°C for 2.5 hr (*Figure 1—figure supplement 2A*) or at 38,000 rpm and 4°C for 4 hr (*Figure 3—figure supplement 1B, C, E, and F* and *Figure 9—figure supplement 2C and D*) by a Himac CP80WX ultracentrifuge (Hitachi) with a P40ST rotor (Hitachi).

To knock down RPL17, HEK293T cells were seeded in complete growth medium in 10 cm$^2$ dishes 24 hr before siRNA transfection. Then, the cells were transfected with RPL17 siRNA (ON-TARGETplus Human RPL17 siRNA, L-013633-01-0005, Horizon Discovery) or control siRNA (ON-TARGETplus Non-targeting Control Pool, D-001810-10-50, Horizon Discovery) by TransIT-X2 Dynamic Delivery System (Mirus) according to the manufacturer's instructions. After 72 hr of siRNA transfection, cells were lysed by the procedure described above.

The gradients were fractionated with continuous measurement of the absorbance at 260 nm by a Triax Flow Cell (BioComp) and Micro collector (ATTO). For ribosomal complex quantification, the straight line connecting the valleys before and after peaks was defined as a background baseline. Then, the area enclosed by the baseline and the peak waveform was calculated. For Western blot (*Figure 2B*) or MS analysis (*Figure 1—figure supplement 2B*), fractions were concentrated using Amicon Ultra filter units (MWCO 10 kDa, Millipore).

## Newly synthesized protein labeling by OP-puro

Metabolic labeling of nascent proteins with OP-puro was performed as previously described (*Iwasaki et al., 2019*). Cells were treated with 20 μM OP-puro and incubated at 37°C for 30 min in a CO$_2$ incubator. After washing with PBS, cells were lysed with buffer containing 20 mM Tris-HCl pH 7.5, 150 mM NaCl, 5 mM MgCl$_2$, and 1% Triton X-100. Nascent polypeptides were labeled with IRdye800CW Azide (LI-COR Biosciences) with a Click-iT Cell Reaction Buffer Kit (Thermo Fisher Scientific). After free dye was removed by a G-25 column (Cytiva), labeled polypeptides were separated by SDS-PAGE. The gel was imaged by Odyssey CLx (LI-COR Biosciences) for the detection of nascent peptides with infrared at 800 nm. Then, total proteins were stained with CBB (FUJIFILM Wako Chemicals) and imaged with an infrared 700 nm signal. The gel area ranging from 17 kDa to 280 kDa was quantified using Image Studio (version 5.2, LI-COR Biosciences), and the nascent peptide signal was normalized to the total protein signal.

## Ribosome profiling

### Library preparation

Ribosome profiling was conducted as previously described (*McGlincy and Ingolia, 2017*; *Mito et al., 2020*). HEK293T cells were cultured in DMEM, high glucose, GlutaMAX Supplement (Thermo Fisher Scientific) supplemented with 10% FBS (Sigma-Aldrich). HAP1 cells were cultured in IMEM (Gibco) supplemented with 10% FBS (Sigma-Aldrich). Both cell lines were maintained at 37°C in a humidified atmosphere containing 5% CO$_2$.

Cell lysates were treated with RNase I, and 17–34-nt protected RNA fragments were gel-excised. After the RNA fragments were ligated with preadenylated linkers, ribosomal RNAs were removed by a Ribo-Zero Gold rRNA Removal Kit (Human/Mouse/Rat) (Illumina, RZG1224). Then, the ligated RNA fragments were reverse-transcribed. The cDNAs were circularized using CircLigaseII (Lucigen) and PCR-amplified. The libraries were sequenced on a HiSeq4000 (Illumina, single reads for 50 nt) for HEK293T cells and a HiSeqX Ten (Illumina, paired ends for 150 nt) for HAP1 cells.

### Data analysis

After removal of the linker sequences, all reads were aligned to human noncoding RNAs (including rRNA, tRNA, snoRNA, snRNA, and microRNA) using STAR (version 2.7.0a) (*Dobin et al., 2013*). Then, the remaining reads were aligned to the human hg38 reference genome by STAR and assigned to canonical transcripts in the University of California, Santa Cruz (UCSC), known gene reference. For HAP1 cell data, the sequences of reads 1 were corrected by reads 2, before quality filtering and adapter sequence trimming.

The A-site offsets for each footprint length were empirically estimated: in the dataset with naïve and *METTL18* KO HEK293T cells, 15 for 20–22, 24, and 28–30-nt-long footprints and 16 for 23 and 32–33-nt-long footprints; in the dataset with *METTL18* KO HEK293T cells with METTL18 WT and

empty vector expression, 15 for 28–31-nt-long footprints and 16 for 32–33-nt-long footprints; and in the dataset with naïve and *METTL18* KO HAP1 cells, 15 for 28–33-nt-long footprints. The reads of each codon were normalized by the average reads per codon of the transcript. We excluded the first and last 5 codons from the analysis. Transcripts with an average reads per codon of 0.3 or higher were considered. The value of ribosome occupancy was the average of normalized reads at each position.

To search for motifs associated with Tyr codon around A-site, ribosome occupancies on the seven amino acid motifs were averaged. Motifs with average ribosome occupancy scores greater than 0.3 were used for the downstream analysis. Then, the $log_2$-fold changes in *METTL18* KO cells over naïve cells were calculated. Motifs with $log_2$-fold changes less than mean –2 SD were subjected to kpLogo (http://kplogo.wi.mit.edu) (*Wu and Bartel, 2017*).

## Northern blot

Total RNA was extracted from cells by TRIzol Reagent (Thermo Fisher Scientific) according to the manufacturer's instructions. Purified RNAs were electrophoresed on Super Sep RNA gels (FUJIFILM Wako Chemicals), transferred onto nylon membranes (Biodyne, Thermo Fisher Scientific), and then UV-crosslinked. The membranes were incubated with UltraHyb-Oligo (Thermo Fisher Scientific) at 37°C for 1 hr. DNA oligonucleotide probes (see below for details) were radiolabeled with [γ−$^{32}$P] ATP (PerkinElmer) by T4 PNK (New England Biolabs) and purified with a G-25 column (Cytiva). After prehybridization, membranes were incubated with a labeled DNA probe overnight at 37°C and washed three times with 2× saline-sodium citrate solution. The signal on the membranes was detected by an Amersham Typhoon (Cytiva) scanner. DNA oligonucleotide sequences used as probes are listed below:

tRNA$^{Tyr}_{GUA}$: 5′-ACAGTCCTCCGCTCTACCAGCTGA-3′, tRNA$^{Leu}_{HAG}$: 5′-CAGCGCCTTAGACCGCT CGGCCA-3′, and U6: 5′-CACGAATTTGCGTGTCATCCTT-3′.

The background-subtracted signal was quantified using ImageQuant TL (Cytiva).

## Hybrid in vitro translation

The hybrid translation assay was performed as described previously (*Erales et al., 2017*; *Panthu et al., 2015*) with optimization for the purpose of this study.

## mRNA preparation

For Fluc reporter, a PCR fragment was PCR-amplified from pGL3 basic (Promega) with the primers 5′-TGACTAATACGACTCACTATAGG-3′ and 5′-TGTATCTTATCATGTCTGCTCGAA-3′. For Rluc-Y0×-Fluc and Rluc-Y39×-Fluc reporters, DNA fragments were PCR-amplified by psiCHECK2-Y0× and Y39× with the primers 5′-TGACTAATACGACTCACTATAGG-3′ and 5′-TGTATCTTATCATGTCTGCTCGAA-3′.

The DNA fragments were used for in vitro transcription by T7-Scribe Standard RNA IVT kit (CELLSCRIPT). Then, the RNAs were capped with the ScriptCap m$^7$G Capping system (CELLSCRIPT) and polyadenylated with the A-Plus poly(A) polymerase Tailing Kit (CELLSCRIPT).

## RRL supernatant

1 ml of RRL (nuclease-treated, Promega) was ultracentrifuged at 240,000 × *g* for 2 hr 15 min at 4°C by an Optima MAX-TL ultracentrifuge (Beckman Coulter) with a TLA110 rotor (Beckman Coulter). Then, 900 µl of supernatant was collected, flash-frozen by liquid nitrogen, and stored at −80°C.

## Ribosome preparation

After brief washing with PBS, cells were resuspended with PBS and centrifuged at 500 × *g* for 3 min. The pellet of cells was resuspended in the same volume of buffer R (10 mM HEPES pH 7.5, 10 mM KOAc, 1 mM MgOAc$_2$, and 1 mM DTT) and incubated for 15 min on ice. Subsequently, the mixture was vortexed for 30 s and centrifuged at 16,000 × *g* for 10 min at 4°C. Then, 300 µl of the supernatant was ultracentrifuged through a 1 ml sucrose cushion (1 M sucrose in buffer R) at 240,000 × *g* for 2 hr 15 min at 4°C by an Optima MAX-TL ultracentrifuge (Beckman Coulter) with a TLA110 rotor (Beckman Coulter). The pellet was rinsed with buffer R2 (20 mM HEPES pH 7.5, 10 mM NaCl, 25 mM KCl, 1.1 mM MgCl$_2$, and 7 mM 2-mercaptoethanol) and then resuspended in 30 µl of buffer R2 by stirring

at 4°C for 15 min, flash-frozen by liquid nitrogen, and stored at −80°C. The ribosome concentration was determined by the absorbance at 260 nm using a NanoDrop (Thermo Fisher Scientific).

## Translation reaction

In vitro translation was typically performed in a 10 µl reaction mixture containing 5 µl of RRL supernatant, 22.6 nM of purified ribosome, 1.1 nM of reporter mRNA, 75 mM KCl, 0.75 mM $MgCl_2$, and 20 µM of amino acid mixture (Promega).

For the Fluc reporter, the reaction mixture was incubated at room temperature for 90 min and stopped by the addition of 20 µl of 1× Passive Lysis Buffer (Promega). The FLuc luminescence from 30 µl of the mixture was measured with the Dual-Luciferase Reporter Assay System (Promega).

For Rluc-Y0×-Fluc and Rluc-Y39×-Fluc reporters, the reaction mixture above was scaled up to 120 µl and incubated at room temperature. 10 µl of the reaction mixture was taken as an aliquot every 5 min and mixed with 20 µl of 1× Passive Lysis Buffer (Promega) to stop the reaction. The Rluc and FLuc luminescence from 30 µl of the mixture was measured with the Dual-Luciferase Reporter Assay System (Promega). The luminescences at 20–30 min were used to calculate the slope of Rluc and Fluc synthesis.

## Proteotoxic-stress reporter assay

pCI-neo Fluc-EGFP (Addgene plasmid #90170; http://n2t.net/addgene:90170; RRID:Addgene_90170) or pCI-neo FlucDM-EGFP (Addgene plasmid #90172; http://n2t.net/addgene:90172; RRID:Addgene_90172) (kind gifts from Franz-Ulrich Hartl) was transfected into naïve cells or *METTL18* KO HEK293T cells by *Trans*IT-293 (Mirus) according to the manufacturer's instructions. For immunofluorescence staining, cells were cultured on a Nunc Lab-Tek II-CC$^2$ Chamber Slide system (Thermo Fisher Scientific) and fixed with 4% paraformaldehyde in PBS for 20 min at room temperature. After being washed twice with PBS containing 0.2% Triton X-100, the cells were incubated with Intercept (TBS) Blocking Buffer (LI-COR Biosciences) with 0.2% Triton X-100 for 1 hr at room temperature. After removal of the blocking buffer, the cells were incubated overnight at 4°C with anti-GFP antibody (Abcam, ab1218) diluted 1:1000 with Intercept (TBS) Blocking Buffer with 0.2% Triton X-100. Then, the cells were washed three times with PBS containing 0.2% Triton X-100 and incubated for 1 hr at room temperature with anti-mouse secondary antibody conjugated with Alexa 488 (Thermo Fisher Scientific, R37120) diluted 1:1000 with Intercept (TBS) Blocking Buffer (LI-COR Biosciences) with 0.2% Triton X-100. Then, the cells were washed three times with PBS containing 0.2% Triton X-100. Slide chambers were mounted with VECTASHIELD HardSet Antifade Mounting Medium (Vector Laboratories). Immunofluorescence images were acquired with a FLUOVIEW FV3000 (Olympus).

To count the aggregation foci, $2 \times 10^5$ cells were seeded in 35-mm glass-bottom dishes (IWAKI) and cultured for 24 hr. Cells were transfected with 2 µg of pCI-neo Fluc-EGFP or pCI-neo FlucDM-EGFP and 4 µg of PEI (Polysciences) and cultured for an additional 24 hr. GFP fluorescence was observed under an FLUOVIEW FV3000 (Olympus). At least 100 GFP-positive cells were observed to count cells with aggregates (n = 3).

For MG132 treatment, cells were treated with 0.25 µM MG132 at 24 hr post transfection. After subsequent 24 hr culture, cells were harvested as described in the 'Ribosome profiling' section.

## Assessment of expression levels of ribosomal RNAs

300 ng of total RNA purified from cell lysates prepared as described in 'Ribosome profiling' section was analyzed with a microchip electrophoresis system (MultiNA, Shimadzu).

## Acknowledgements

We are grateful to all the members of the Iwasaki, Shinkai, Ito, Dohmae, and Sodeoka laboratories for constructive discussion, technical help, and critical reading of the manuscript. This study was supported in mass spectrometry, Sanger sequencing, and confocal microscopy by the Support Unit for Bio-Material Analysis from RIKEN CBS Research Resources Division and in the use of the supercomputer HOKUSAI SailingShip by RIKEN ACCC. This study used the following kindly provided materials: PX330-B/B from Tetsuro Hirose, pL-CRISPR.EFS.tRFP from Benjamin Ebert, pCI-neo Fluc-EGFP and pCI-neo FlucDM-EGFP from Franz-Ulrich Hartl, and a plasmid for *Salmonella* MTAN expression from Vern Schramm. SI was supported by a Grant-in-Aid for Transformative Research Areas (B)

'Parametric Translation' (JP20H05784) from the Ministry of Education, Culture, Sports, Science and Technology (MEXT), a Grant-in-Aid for Young Scientists (A) (JP17H04998) and a Challenging Research (Exploratory) (JP19K22406) from the Japan Society for the Promotion of Science (JSPS), AMED-CREST (JP21gm1410001) from the Japan Agency for Medical Research and Development (AMED), and the Pioneering project ('Biology of Intracellular Environments') and Aging Project from RIKEN. Y Shinkai was supported by a Grant-in-Aid for Scientific Research (A) (JP18H03991) and Scientific Research on Innovative Areas 'Chromatin potential for gene regulation' (JP18H05530) from JSPS and the Pioneering project ('Epigenome manipulation') from RIKEN. EMS was supported by a Grant-in-Aid for Scientific Research (C) (JP21K06026) from JPSP and the Collaboration seed fund and Incentive Research Project from RIKEN. TS was supported by a Grant-in-Aid for Scientific Research (C) (JP20K06497) from JSPS. TI was supported by a Grant-in-Aid for Scientific Research (B) (JP19H03172) by JSPS, a Grant-in-Aid for Transformative Research Areas (A) 'Biology of Non-domain Biopolymer' (JP21H05281) by MEXT, the BDR Structural Cell Biology Project, the Pioneering Projects ('Dynamic Structural Biology' and 'Biology of Intracellular Environments'') and Aging Project from RIKEN, and AMED-CREST (JP21gm1410001) from AMED. DNA libraries were sequenced by the Vincent J Coates Genomics Sequencing Laboratory at UC Berkeley, supported by NIH S10 OD018174 Instrumentation Grant. Structural analysis was supported by the Platform Project for Supporting Drug Discovery and Life Science Research (Basis for Supporting Innovative Drug Discovery and Life Science Research [BINDS], JP21am0101082) from AMED.

## Additional information

### Funding

| Funder | Grant reference number | Author |
| --- | --- | --- |
| RIKEN | Pioneering project | Shintaro Iwasaki<br>Yoichi Shinkai<br>Takuhiro Ito |
| RIKEN | Aging Project | Shintaro Iwasaki<br>Takuhiro Ito |
| RIKEN | Collaboration seed fund | Eriko Matsuura-Suzuki |
| RIKEN | Incentive Research Project | Eriko Matsuura-Suzuki |
| RIKEN | BDR Structural Cell Biology Project | Takuhiro Ito |
| Ministry of Education, Culture, Sports, Science and Technology | JP20H05784 | Shintaro Iwasaki |
| Japan Society for the Promotion of Science | JP17H04998 | Shintaro Iwasaki |
| Japan Society for the Promotion of Science | JP19K22406 | Shintaro Iwasaki |
| Japan Agency for Medical Research and Development | JP21gm1410001 | Shintaro Iwasaki<br>Takuhiro Ito |
| Japan Society for the Promotion of Science | JP18H03991 | Yoichi Shinkai |
| Japan Society for the Promotion of Science | JP18H05530 | Yoichi Shinkai |
| Japan Society for the Promotion of Science | JP21K06026 | Eriko Matsuura-Suzuki |
| Japan Agency for Medical Research and Development | JP21am0101082 | Takuhiro Ito |

| Funder | Grant reference number | Author |
| --- | --- | --- |
| Japan Society for the Promotion of Science | JP20K06497 | Tadahiro Shimazu |
| Japan Society for the Promotion of Science | JP19H03172 | Takuhiro Ito |
| Ministry of Education, Culture, Sports, Science and Technology | JP21H05281 | Takuhiro Ito |

The funders had no role in study design, data collection and interpretation, or the decision to submit the work for publication.

## Author contributions

Eriko Matsuura-Suzuki, Conceptualization, Formal analysis, Funding acquisition, Investigation, Methodology, Visualization, Writing – review and editing; Tadahiro Shimazu, Conceptualization, Formal analysis, Funding acquisition, Investigation, Methodology, Supervision, Visualization, Writing – review and editing; Mari Takahashi, Kaoru Kotoshiba, Takehiro Suzuki, Kazuhiro Kashiwagi, Formal analysis, Investigation, Writing – review and editing; Yoshihiro Sohtome, Mai Akakabe, Resources, Writing – review and editing; Mikiko Sodeoka, Resources, Supervision, Writing – review and editing; Naoshi Dohmae, Supervision, Writing – review and editing; Takuhiro Ito, Conceptualization, Funding acquisition, Methodology, Supervision, Visualization, Writing – review and editing; Yoichi Shinkai, Conceptualization, Funding acquisition, Methodology, Supervision, Writing – review and editing; Shintaro Iwasaki, Conceptualization, Formal analysis, Funding acquisition, Methodology, Supervision, Visualization, Writing – original draft, Writing – review and editing

## Author ORCIDs

Eriko Matsuura-Suzuki http://orcid.org/0000-0002-1485-7959
Tadahiro Shimazu http://orcid.org/0000-0003-2331-091X
Mari Takahashi http://orcid.org/0000-0002-9995-0019
Kazuhiro Kashiwagi http://orcid.org/0000-0001-6470-5817
Yoshihiro Sohtome http://orcid.org/0000-0002-9165-6720
Mikiko Sodeoka http://orcid.org/0000-0002-1344-364X
Naoshi Dohmae http://orcid.org/0000-0002-5242-9410
Takuhiro Ito http://orcid.org/0000-0003-3704-5205
Yoichi Shinkai http://orcid.org/0000-0002-6051-2484
Shintaro Iwasaki http://orcid.org/0000-0001-7724-3754

## Decision letter and Author response

Decision letter https://doi.org/10.7554/eLife.72780.sa1
Author response https://doi.org/10.7554/eLife.72780.sa2

## Additional files

### Supplementary files

• Transparent reporting form

### Data availability

The ribosome profiling data obtained in this study (GSE179854 and GSE200172) were deposited to the National Center for Biotechnology Information (NCBI). The proteome data for ProSeAM SILAC (ID: PXD026813) were available via ProteomeXchange. The structural coordinates (PDB: 7F5S) and cryo-EM maps (EMDB: EMD-31465) of His245 methylation-deficient ribosomes have been deposited in the Protein Data Bank (PDB) and Electron Microscopy Data Bank (EMDB), respectively.

The following datasets were generated:

| Author(s) | Year | Dataset title | Dataset URL | Database and Identifier |
|---|---|---|---|---|
| Eriko MS | 2021 | METTL18-mediated histidine methylation on RPL3 modulates translation elongation for proteostasis maintenance | http://www.ncbi.nlm.nih.gov/geo/query/acc.cgi?acc=GSE179854 | NCBI Gene Expression Omnibus, GSE179854 |
| Tadahiro S | 2021 | ProSeAM SILAC MS screening for METTL18 substrates | https://repository.jpostdb.org/preview/27012566060cfe86cb2c91 | jPOST repository, JPST001223 |
| Mari T, Kazuhiro K, Takuhiro I | 2021 | human delta-METTL18 60S ribosome | https://www.rcsb.org/structure/7F5S | RCSB Protein Data Bank, 7F5S |
| Eriko MS | 2022 | METTL18-mediated histidine methylation on RPL3 modulates translation elongation for proteostasis maintenance | http://www.ncbi.nlm.nih.gov/geo/query/acc.cgi?acc=GSE200172 | NCBI Gene Expression Omnibus, GSE200172 |

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

## Appendix 1

### Appendix 1—key resources table

| Reagent type (species) or resource | Designation | Source or reference | Identifiers | Additional information |
|---|---|---|---|---|
| Gene (*Homo sapiens*) | *METTL18* | GenBank | NM_033418 | |
| Gene (*H. sapiens*) | *RPL3* | GenBank | NM_000967 | |
| Gene (*Mus musculus*) | *METTL18* | GenBank | NM_027279 | cDNA clone AK139786 (FANTOM) was used |
| Gene (*M. musculus*) | *RPL3* | GenBank | NM_013762 | |
| Cell line (*H. sapiens*) | Naïve HEK293T | RIKEN BRC | RCB2202 | Female |
| Cell line (*H. sapiens*) | *METTL18* KO HEK293T | This paper | | Female; CRISPR/Cas9-edited cell line, knocking out *METTL18* |
| Cell line (*H. sapiens*) | *METTL18* KO HEK293T with stable METTL18 expression | This paper | | Female; exogenous METTL18 expression was induced in *METTL18* KO HEK293T |
| Cell line (*H. sapiens*) | *SETD3* KO HEK293T | This paper | | Female; CRISPR/Cas9-edited cell line, knocking out *SETD3* |
| Cell line (*H. sapiens*) | *SETD3*-METTL18 DKO HEK293T | This paper | | Female; CRISPR/Cas9-edited cell line, knocking out *SETD3 and METTL18* simultaneously |
| Cell line (*H. sapiens*) | Naïve HAP1 | Horizon Discovery | Cat# C631 | Male |
| Cell line (*H. sapiens*) | HAP1 1-nt deletion | Horizon Discovery | Cat# HZGHC000541c009 | Male; CRISPR/Cas9-edited cell line containing a 1-nt deletion in a coding exon of METTL18 |
| Cell line (*H. sapiens*) | HAP1 2-nt deletion | Horizon Discovery | Cat# HZGHC000541c012 | Male; CRISPR/Cas9-edited cell line containing a 2-nt deletion in a coding exon of METTL18 |
| Cell line (*H. sapiens*) | HAP1 4-nt deletion | Horizon Discovery | Cat# HZGHC000541c002 | Male; CRISPR/Cas9-edited cell line containing a 4-nt deletion in a coding exon of METTL18 |
| Transfected construct (*H. sapiens*) | PX330-B/B-gMETTL18 | This paper | | Guide RNA expression |
| Transfected construct (*H. sapiens*) | pL-CRISPR.EFS.tRFP-gSETD3 | This paper | | Guide RNA expression |
| Transfected construct (*H. sapiens*) | pcDNA3-hRPL3-FLAG (WT and His245Ala) | This paper | | Protein expression |
| Transfected construct (*H. sapiens*) | pcDNA3-mRPL3-FLAG (WT and His245Ala) | This paper | | Protein expression |
| Transfected construct (*H. sapiens*) | pQCXIP-hMETTL18-HA | This paper | | Protein expression |
| Transfected construct (*H. sapiens*) | hMETTL18-Asp193Lys-Gly195Arg-Gly197Arg-HA | This paper | | Protein expression |
| Transfected construct (*H. sapiens*) | siRNA to RPL17 | Horizon Discovery | L-013633-01-0005 | |
| Transfected construct (*H. sapiens*) | Control siRNA | Horizon Discovery | D-001810-10-50 | |
| Transfected construct (*H. sapiens*) | pCI-neo Fluc-EGFP | Addgene | RRID:Addgene_90170 | Protein expression |

*Appendix 1 Continued on next page*

*Appendix 1 Continued*

| Reagent type (species) or resource | Designation | Source or reference | Identifiers | Additional information |
|---|---|---|---|---|
| Transfected construct (*H. sapiens*) | pCI-neo FlucDM-EGFP | Addgene | RRID:Addgene_90172 | Protein expression |
| Antibody | Anti-α-tubulin (mouse monoclonal) | Sigma-Aldrich | Cat# T5168; RRID:AB_477579 | WB 1:1000 |
| Antibody | Anti-METTL18 (rabbit polyclonal) | Proteintech Group | Cat# 25553-1-AP; RRID:AB_2503968 | WB (1:1000) |
| Antibody | Anti-SETD3 (rabbit polyclonal) | Abcam | Cat# ab174662; RRID:AB_2750852 | WB (1:1000) |
| Antibody | Anti-RPL3 (mouse monoclonal) | Proteintech Group | Cat# 66130-1-Ig; RRID:AB_2881529 | WB (1:1000) |
| Antibody | Anti-RPL3 (rabbit polyclonal) | Proteintech Group | Cat# 11005-1-AP; RRID:AB_2181760 | WB (1:1000) |
| Antibody | Anti-PES1 (rat monoclonal) | Abcam | Cat# ab252849; RRID:AB_2915993 | WB (1:1000) |
| Antibody | Anti-NMD3 (rabbit monoclonal) | Abcam | Cat# ab170898; RRID:AB_2915994 | WB (1:1000) |
| Antibody | Anti-HA (mouse monoclonal) | MBL | Cat# M180-3; RRID:AB_10951811 | WB (1:1000) |
| Antibody | Anti-GFP (rabbit polyclonal) | Abcam | Cat# ab6556; RRID:AB_305564 | WB (1:1000) |
| Antibody | Anti-β-actin (mouse monoclonal) | MBL | Cat# M177-3; RRID:AB_10697039 | WB (1:1000) |
| Antibody | Anti-mouse IgG, conjugate with HRP (sheep polyclonal) | Cytiva | Cat# NA931V; RRID:AB_772210 | WB (1:5000) |
| Antibody | Anti-rabbit IgG, conjugated with HRP (donkey polyclonal) | Cytiva | Cat# NA934V; RRID:AB_772206 | WB (1:5000) |
| Antibody | Anti-mouse IgG, conjugated with IRDye 680RD (goat polyclonal) | LI-COR Biosciences | Cat# 925-68070; RRID:AB_2651128 | WB (1:10,000) |
| Antibody | Anti-rabbit IgG, conjugated with IRDye 680RD (goat polyclonal) | LI-COR Biosciences | Cat# 925-68071; RRID:AB_2721181 | WB (1:10,000) |
| Antibody | Anti-mouse IgG, conjugated with IRDye 800CW (goat polyclonal) | LI-COR Biosciences | Cat# 926-32210; RRID:AB_621842 | WB (1:10,000) |
| Antibody | Anti-rabbit IgG, conjugated with IRDye 800CW (goat polyclonal) | LI-COR Biosciences | Cat# 926-32211; RRID:AB_621843 | WB (1:10,000) |
| Antibody | Anti-rat IgG, conjugated with IRDye 800CW (goat polyclonal) | LI-COR Biosciences | Cat# 926-32219; RRID:AB_1850025 | WB (1:10,000) |
| Antibody | Anti-GFP (mouse monoclonal) | Abcam | Cat# ab1218; RRID:AB_298911 | IF (1:1000) |
| Antibody | Anti-mouse IgG, conjugated with Alexa Fluor 488 (goat polyclonal) | Thermo Fisher Scientific | Cat# R37120; RRID:AB_2556548 | IF (1:1000) |
| Recombinant DNA reagent | pET19b-mMETTL18 | This paper | | Expression of N-terminally His-tagged mouse METTL18 in *Escherichia coli* |
| Recombinant DNA reagent | pCold-GST-mMETTL18 | This paper | | Expression of N-terminally His- and GST-tagged mouse METTL18 in *E. coli* |

*Appendix 1 Continued on next page*

*Appendix 1 Continued*

| Reagent type (species) or resource | Designation | Source or reference | Identifiers | Additional information |
|---|---|---|---|---|
| Recombinant DNA reagent | *Salmonella* MTAN | Addgene | RRID:Addgene_64041 | Expression of *Salmonella* MTAN in *E. coli* |
| Recombinant DNA reagent | pGL3 basic | Promega | Cat# E1751 | |
| Recombinant DNA reagent | psiCHECK2 | Promega | Cat# C8021 | |
| Recombinant DNA reagent | psiCHECK2-Y0× | This study | | Encoding Rluc-Fluc fusion |
| Recombinant DNA reagent | psiCHECK2-Y39× | This study | | Encoding Rluc-Fluc fusion with Tyr repeat insertion |
| Sequence-based reagent | Probe for tRNA$^{Tyr}_{GUA}$ | This paper | | 5'-ACAGTCCTCCGCTCTACCAGCTGA-3' |
| Sequence-based reagent | Probe for tRNA$^{Leu}_{HAG}$ | This paper | | 5'-CAGCGCCTTAGACCGCTCGGCCA-3' |
| Sequence-based reagent | Probe for U6 | This paper | | 5'-CACGAATTTGCGTGTCATCCTT-3' |
| Commercial assay or kit | QuikChange Site-Directed Mutagenesis Kit | Agilent Technologies | Cat# 200518 | |
| Commercial assay or kit | PEI transfection reagent | Polysciences | | |
| Commercial assay or kit | TransIT-293 | Mirus | Cat# MIR2700 | |
| Commercial assay or kit | TransIT-X2 Dynamic Delivery System | Mirus | Cat# MIR6000 | |
| Commercial assay or kit | Dual-Luciferase Reporter Assay System | Promega | Cat# E1910 | |
| Commercial assay or kit | Rabbit Reticulocyte Lysate, Nuclease-Treated | Promega | Cat# L4960 | |
| Commercial assay or kit | Click-iT Cell Reaction Buffer Kit | Thermo Fisher Scientific | Cat# C10269 | |
| Commercial assay or kit | T7-Scribe Standard RNA IVT kit | CELLSCRIPT | Cat# C-MSC11610 | |
| Commercial assay or kit | ScriptCap m$^7$G Capping system | CELLSCRIPT | Cat# C-SCCE0625 | |
| Commercial assay or kit | A-Plus poly(A) polymerase Tailing kit | CELLSCRIPT | Cat# C-PAP5104H | |
| Chemical compound, drug | IRdye800CW azide | LI-COR Biosciences | Cat# 929-65000 | |
| Chemical compound, drug | MG132 | FUJIFILM Wako Chemicals | Cat# 139-18451 | |
| Software, algorithm | Proteome Discoverer | Thermo Fisher Scientific | Version 2.3 | LC-MS/MS for methylated peptide |
| Software, algorithm | Proteome Discoverer | Thermo Fisher Scientific | Version 2.4 | SILAC-MS |
| Software, algorithm | MASCOT | Matrix Science | Version 2.7 | LC-MS/MS for methylated peptide and SILAC-MS |
| Software, algorithm | RELION-3.1 | https://doi.org/10.1107/S2052252520000081 | | |
| Software, algorithm | CTFFIND-4.1 | https://doi.org/10.1016/j.jsb.2015.08.008 | | |

*Appendix 1 Continued on next page*

*Appendix 1 Continued*

| Reagent type (species) or resource | Designation | Source or reference | Identifiers | Additional information |
|---|---|---|---|---|
| Software, algorithm | PHENIX | https://doi.org/10.1107/S0907444909052925 | | |
| Software, algorithm | Coot | https://doi.org/10.1107/S0907444910007493 | | |
| Software, algorithm | Image Studio | LI-COR Biosciences | Version 5.2 | |
| Software, algorithm | STAR | https://doi.org/10.1093/bioinformatics/bts635 | Version 2.7.0a | |
| Software, algorithm | kpLog | https://doi.org/10.1093/nar/gkx323 | http://kplogo.wi.mit.edu | |

