## [Editor Report]

This study investigates METTLL18-mediated RPL3 histidine methylation on 245 position and how it regulates translation elongation and protects cells from cellular aggregation of Tyr-rich proteins. The study potentially provides some new example of ‘ribosome code’ and how ribosome PTM could affect protein translation.

---

## [Decision Letter]

**Decision letter after peer review:**

Thank you for submitting your article "METTL18-mediated histidine methylation on RPL3 modulates translation elongation for proteostasis maintenance" for consideration by *eLife*. Your article has been reviewed by 3 peer reviewers, including Qing Zhang as the Reviewing Editor and Reviewer #1, and the evaluation has been overseen by Volker Dötsch as the Senior Editor. The following individual involved in review of your submission has agreed to reveal their identity: Sarah E Walker (Reviewer #3).

Essential revisions:

1) Experimentally address the discrepancy between the current study and recently published study.

2) In addition, authors need to examine whether the phenotype of METTL18KO is mediated through change with RPL3 methylation as well as Tyr codons/faster peptidyl transfer.

3) More evidence is needed for claiming strongly that methylation slows translocation specifically. If not, the statement may need to tune down. Also, the claims with regarding to ribosome code or translation tuning need to tune down.

*Reviewer #2 (Recommendations for the authors):*

It is my opinion that authors experiments and data on Tyrosine specific effects and proteostatsis maintenance are not fully supported by the current experimental data. These would be the major novelty in the manuscript in the comparison to the previously published work by Malecki et al., 2021. There are several reasons for my concerns:

1. The effects on Tyr codons are seen in HEK293 cells but not in HAP1 cells (used in work by Malecki et al., 2021) even though the experimental approach (ribosome profiling) and genetic manipulation (CRISPR/Cas9 KO of METTL18) is done in the same/similar way. Assuming that authors analyzed data from the previously published study (Malecki et al., 2021) and did not find effects on Tyr codons would authors argue for the cell specific effects of RPL3 His245 methylation?

2. Current explanation of the aggregation data is purely speculative. Protein folding is a complex process and proteins with less tyrosine residues would be affected by this mechanism based on the position of tyrosine residues in the folding core of the particular protein. If the authors argument is that translation rate impacts overall protein folding due to the number of Tyr residues in protein. One would expect to see drastic change in translation rate and abundance of certain proteins. Is this indeed the case?

3. In respect to the point mentioned in 2. MACROH2A1is a protein with 8 Tyr codons and 2 depicted in the figure 5H are found in the C-termini of the protein not influencing either histone or macro domain folding or structure. Are other Tyr codons in this protein impacted by METTL18 KO?

4. If the effect of protein aggregation in METTL18 KO is dependent solely on Tyr codons and the faster peptidyl transfer (not decoding) would depleting pull of charged Tyr tRNAs and not tRNA abundance (as authors tested) be a good control. One could target tyrosyl-tRNA synthetase by siRNA or shRNA and reduce protein aggregation in METTL18 KO?

*Reviewer #3 (Recommendations for the authors):*

Please add number of replicates for experiments to the methods and/or figure legends.

Please add antibody dilutions used to the methods section.

---

## [Author Response]

Essential revisions:1) Experimentally address the discrepancy between the current study and recently published study.

As recommended, we conducted ribosome profiling of the HAP1 cells with *METTL18* KO. For this assay, we employed 3 cell lines, including the same cell line used in an earlier report (Małecki *et al. NAR* 2021). Indeed, we observed similarly enhanced elongation on Tyr codons (*i.e.*, reduction of footprints on the codons) in the *METTL18* KO HAP1 cell lines as in the *METTL18* KO HEK293T cells that we originally used. Although we still do not have any explanation for the difference between our data and the published ribosome profiling data, the new data supported our conclusion that RPL3 methylation exerts amino acid-specific effects on elongation.

2) In addition, authors need to examine whether the phenotype of METTL18KO is mediated through change with RPL3 methylation as well as Tyr codons/faster peptidyl transfer.

To directly address the effect of RPL3 methylation on Tyr codon elongation, we harnessed the *Renilla*–firefly luciferase fusion reporter system (Kisly *et al. NAR* 2021) and hybrid in vitro translation (Panthu *et al. Biochem J* 2015 and Erales *et al. PNAS* 2017). Using this setup, we observed that ribosomes isolated from *METTL18* KO cells perform faster elongation of Tyr codons. Given the direct comparison of ribosomes with and without RPL3 methylation, these data provide solid evidence of a relationship between RPL3 methylation and translation elongation at Tyr codons.

3) More evidence is needed for claiming strongly that methylation slows translocation specifically. If not, the statement may need to tune down. Also, the claims with regarding to ribosome code or translation tuning need to tune down.

Although the in vitro translation mentioned above showed direct evidence that Tyr codon elongation is retarded by methylated RPL3, understanding, the specificity in more detail would require further experiments. Thus, we modified our description of the data to avoid claims of pinpointing the affected step in elongation processes (such as peptidyl transfer or translocation) throughout the manuscript. Additionally, we avoided using “ribosome code” and “translation tuning” in the manuscript.

Reviewer #2 (Recommendations for the authors):It is my opinion that authors experiments and data on Tyrosine specific effects and proteostatsis maintenance are not fully supported by the current experimental data. These would be the major novelty in the manuscript in the comparison to the previously published work by Malecki et al., 2021. There are several reasons for my concerns:1. The effects on Tyr codons are seen in HEK293 cells but not in HAP1 cells (used in work by Malecki et al., 2021) even though the experimental approach (ribosome profiling) and genetic manipulation (CRISPR/Cas9 KO of METTL18) is done in the same/similar way. Assuming that authors analyzed data from the previously published study (Malecki et al., 2021) and did not find effects on Tyr codons would authors argue for the cell specific effects of RPL3 His245 methylation?

As suggested, we performed ribosome profiling on 3 independent KO lines in HAP1 cells, including the one used in the NAR paper (*METTL18* KO, 2-nt del.). Indeed, all *METTL18* KO HAP1 cells showed a reduction in footprints at Tyr codons, as observed in HEK293 cells (see Author response image 1 and Figure 4H), and thus, there was a consistent effect of RPL3 methylation on elongation irrespective of the cell type. On the other hand, we could not find such a trend (see Author response image 1) by reanalysis of the published data (Małecki *et al. NAR* 2021).

**Author response image 1. sa2fig1:** 

Thus far, we could not find the origin of the difference in ribosome profiling compared to the earlier paper. Culturing conditions or others may affect the data. Given that, we amended the discussion to cover the potential of context/situation-dependent effects of RPL3 methylation.

2. Current explanation of the aggregation data is purely speculative. Protein folding is a complex process and proteins with less tyrosine residues would be affected by this mechanism based on the position of tyrosine residues in the folding core of the particular protein. If the authors argument is that translation rate impacts overall protein folding due to the number of Tyr residues in protein. One would expect to see drastic change in translation rate and abundance of certain proteins. Is this indeed the case?

We thank the reviewer for the constructive suggestion. Here, we performed additional SILAC experiments to assess the total proteome alterations (see Figure 8). To evaluate the fraction of proteins degraded, we combined this approach with treatment with MG132, a proteasome inhibitor (Figure 8A). This approach indeed revealed a subset of proteins that were actively broken down in *METTL18* KO cells (Figure 8B); the amount of these proteins was reduced in *METTL18* KO cells and was recovered by MG132 treatment. This group of proteins included those enriched with Tyr residues (Figure 8C). Thus, enhanced elongation of Tyr residues leads to proteins being susceptible to proteasomal degradation. In other words, RPL3 methylation by METTL18 maintains the proteome integrity from such an imbalance between the synthesis and degradation of proteins.

3. In respect to the point mentioned in 2. MACROH2A1is a protein with 8 Tyr codons and 2 depicted in the figure 5H are found in the C-termini of the protein not influencing either histone or macro domain folding or structure. Are other Tyr codons in this protein impacted by METTL18 KO?

Accordingly, we depicted the whole CDS of MACROH2A1 (Figure 7 — figure supplement 1C) for footprint read accumulation and aggregation percentage provided by TANGO. MACROH2A1 has 7 Tyr codons, and 3 of them overlap with aggregation-prone regions (amino acid positions at 47, 336, and 349). In addition to 336 and 349 highlighted in Figure 7E, we prepared an additional zoomed-up panel for position 47 in Figure 7 — figure supplement 1D. All 3 positions were accompanied by reduced footprints with METTL18 depletion.

4. If the effect of protein aggregation in METTL18 KO is dependent solely on Tyr codons and the faster peptidyl transfer (not decoding) would depleting pull of charged Tyr tRNAs and not tRNA abundance (as authors tested) be a good control. One could target tyrosyl-tRNA synthetase by siRNA or shRNA and reduce protein aggregation in METTL18 KO?

We truly appreciate this constructive suggestion. We did indeed try knockdown of tyrosyl-tRNA synthetase by siRNA in HEK293T cells with FlucDM aggregation reporter expression. However, we noted that siRNA transfection *per se* was quite stressful to cells so that FlucDM was highly aggregated even in the naïve HEK293T cells (see Author response image 2), comparable to untransfected *METTL18* KO cells (see Figure 6B). Therefore, the suggested experiments were unfortunately challenging. Instead, we modified the statement in the manuscript and avoided claiming to have pinpointed the step (*i.e.*, peptidyl transfer or translocation) that RPL3 methylation affects.

Reviewer #3 (Recommendations for the authors):Please add number of replicates for experiments to the methods and/or figure legends.

According to the reviewer's suggestion, we amended the figure legends to clearly describe the replicate numbers.

Please add antibody dilutions used to the methods section.

According to the reviewer's suggestion, we added the corresponding information to the methods section and key resource table.